# Understanding the effects of stress on the P300 response during naturalistic simulation of heights exposure

**Howe Yuan Zhu**[1]*, **Hsiang-Ting Chen**[2], **Chin-Teng Lin**[1]

**1** Australian AI Institute, GrapheneX-UTS Human-centric AI Centre, University of Technology Sydney, Sydney, New South Wales, Australia, **2** School of Computer Science, University of Adelaide, Adelaide, South Australia, Australia

* Howe.Zhu@uts.edu.au

**Data Availability Statement:** All data required to produce the results in the paper are available at: https://datadryad.org/stash/share/tRM7LgrL_ecB9ghA632P_FXbMIYN6Yo8121G1f36mak.

## Abstract

Stress is a prevalent bodily response universally experienced and significantly affects a person's mental and cognitive state. The P300 response is a commonly observed brain behaviour that provides insight into a person's cognitive state. Previous works have documented the effects of stress on the P300 behaviour; however, only a few have explored the performance in a mobile and naturalistic experimental setup. Our study examined the effects of stress on the human brain's P300 behaviour through a height exposure experiment that incorporates complex visual, vestibular, and proprioceptive stimuli. A more complex sensory environment could produce translatable findings toward real-world behaviour and benefit emerging technologies such as brain-computer interfaces. Seventeen participants experienced our experiment that elicited the stress response through physical and virtual height exposure. We found two unique groups within our participants that exhibited contrasting behavioural performance and P300 target reaction response when exposed to stressors (from walking at heights). One group performed worse when exposed to heights and exhibited a significant decrease in parietal P300 peak amplitude and increased beta and gamma power. On the other hand, the group less affected by stress exhibited a change in their N170 peak amplitude and alpha/mu rhythm desynchronisation. The findings of our study suggest that a more individualised approach to assessing a person's behaviour performance under stress can aid in understanding P300 performance when experiencing stress.

## Introduction

The human stress response is an essential survival mechanism closely tied to our ability to perceive and react to potential oncoming danger [1, 2]. The homeostatic balance of stress levels can significantly impact a person's cognitive and physical abilities [3, 4]. Stress is characterised and modeled by the Russell-circumplex model as an increase in arousal (cognitive activity) and a negative valence (unpleasant/unhappy state) [5, 6]. Past researchers have theorised the relationship between stress on the brain's cognitive ability and performance. Some studies

**Funding:** This work was supported by Australian Research Council (ARC, https://www.arc.gov.au/) (DP210101093 and DP220100803), Australia Defence Innovation Hub (https://innovationhub. defence.gov.au/, P18-650825), US Office of Naval Research Global (https://www.nre.navy.mil/, ONRG - NICOP - N62909-19-1-2058), AFOSR – DST Australia Autonomy Initiative (ID10134), and NSW Defence Innovation Network (https:// innovationhub.defence.gov.au/, DINPP2019 S1-03/ 09 and PP21-22.03.02) to Chin-Teng Lin. The funders had no role in study design, data collection and analysis, decision to publish, or preparation of the manuscript.

**Competing interests:** The authors have declared that no competing interests exist.

[7–9] suggested that the bodily stress response negatively affects the brain's cognitive functions. These studies found that the stress-induced glucocorticoid release and catecholamine effects can hinder or impair decision-making [7] and memory recollection [8]. Conversely, other studies [10–13] have found that stress enhances cognitive performance. These studies found a correlation between stress-induced glucocorticoid release and improved cognitive behaviours such as decision-making [13] and dual-tasking performance [11]. These works yield seemingly contradictory results, yet both findings seem intuitively plausible. Other explanations for the variance could be differences between sample groups, study methodologies, and the effectiveness of the stress elicitation paradigm. This factor could have caused the varying findings. Another popular theory is Yerkes-Dodson (YD) law [14, 15]. The YD law dictates that stress/arousal has a parabolic relation to cognitive performance. A person will experience low performance at either extremity at lower (bored or drowsy) and higher (panic and intense anxiety) arousal levels while having an optimal performance at a middle arousal level [15]. A person's arousal level has been found to be linked to a cognitive response to urgent and sudden events such as threat perception [16].

Researchers have further explored the effects of stress on cognitive function by measuring the brain's P300 response using Electroencephalograph (EEG). The P300 or P3 wave is an Event-related potential (ERP) component indicated through parietocentral positivity when measured through EEG [17]. Brain-computer interface (BCI) systems commonly choose the P300 paradigm because it is highly reliable, with flexible stimuli and a distinct signal peak for classification [18, 19]. Previous studies that explored the relationship between stress and the P300 response have also found varying results for the effect of stress on the P300 amplitude Table 1. Certain studies have reported a decrease in ERP P300 amplitude for multiple different paradigms such as oddball [20–22], Go/No-Go [23], Dot Probe task [3], and Flanker task [24]. However, other studies have found varying effects with stress, either increasing [25, 26] or producing no change [27] in P300 amplitude. Kamp et al. [20] rationalised this contradiction by questioning the effectiveness of eliciting the stress response. Their study ultimately found that P300 amplitude decreases during an Oddball task when stress is induced through the Trier Social Stress Test (TSST). Thus the effects of stress on the P300 response still need to be clarified.

Traditional EEG research tends to minimise EEG signal noise through subjects maintaining a stationary seated or standing position with minimal background auditory noise, constant lighting, and consistent emotional behaviour (baseline to calm subjects) [28]. The shift between the real world and the laboratory is a dramatic multifactorial change in the paradigm. One commonality observed in the previous studies is the use of stationary means for eliciting the stress response, such as Stroop Tasks [27], Paced Auditory Serial Addition Test (PASAT)

**Table 1. A table outlining the varying results of different studies and the conflicting results for the relation between P300 amplitude and stress.**

| Ref | Stressor task | P300 Task | Sample Size | Stress Effect |
|---|---|---|---|---|
| Kamp et al. (2021) [20] | TSST | Oddball | 63 | ↓ P300 Amplitude |
| Ceballos et al. (2012) [21] | PASAT | Oddball | 75 | ↓ P300 Amplitude |
| Granovsky et al. (1998) [22] | Word Triggers | Oddball | 14 | ↓ P300 Amplitude |
| Dierolf et al. (2017) [23] | SECPT | Go/NoGo | 39 | ↓ P300 Amplitude |
| Jiang et al. (2017) [3] | TSST | Dot Probe Task | 62 | ↓ P300 Amplitude |
| Mingming et al. (2018) [24] | Mental Arithmetic Task | Flanker Task | 17 | ↓ P300 Amplitude |
| Dierolf et al. (2018) [25] | TSST | Go/NoGo | 49 | ↑ P300 Amplitude |
| Mingming et al. (2020) [26] | Mental Arithmetic Task | Flanker Task | 20 | ↑ P300 Amplitude |
| Garcia et al. (2019) [27] | Stroop Task | BCI Speller | 7 | - P300 Amplitude |

[21], Socially Evaluated Cold Pressor Task (SECPT) [23], Mental Arithmetic Task [24, 26] and TSST [3, 20]. While these methods are effective, they lack real-world factors such as complex visual environments, vestibular stimuli, and proprioceptive feedback.

Our work aims to investigate further the relationship between stress (the perception of threat) and P300 response in a mobile naturalistic experiment environment. Our studies differ from previous works using an ambulatory stress elicitation paradigm. Therefore, we selected walking at heights/acrophobia as the stress elicitation method. Acrophobia is a multisensory approach to eliciting a stress response that has been consistent across age and demographics [29, 30]. Inducing stress through walking at heights provides more realistic and translatable research to real-world behaviour. This height experiment design leverages the advantages of head-mounted Virtual Reality (VR) [31, 32], virtual height environment [4], immersion background noise [33–35], and a novel physically elevated platform (see Fig 1) [36, 37].

Our experiment consists of two 2.4m physical walking platforms: a platform on the ground (0.02m elevation) and a physically elevated platform (0.65m elevation). Within virtual reality, the participant will experience three different virtual heights, on the ground (0.02m), slightly elevated (0.65m), and at extreme virtual height (150m). 17 participants experienced four experimental conditions (see Fig 1): Ground-Ground (GG) -physically on the ground and virtually on the ground, Platform-Ground (PG) -physically on the elevated platform and virtually on the ground, Platform-Platform (PP)—physically on the elevated platform and virtually on the elevated platform, and Platform-Height (PH)- physically on the elevated platform and virtually on an extremely height level. These four conditions were designed to form four levels of stress, with GG being the lower extremity, PH the upper extremity, and PG/PP being the middle levels. Participants would perform four walks on each condition and perform twenty-five Oddball (with haptic reaction input) trials between each walk (see Fig 2). The participants wore a wireless high-density 64-channels EEG scalp cap and performed the elevated walking trials with an attached safety harness. We compared each condition to measure the self-assessment manikin questionnaire responses, target stimuli reaction time, and EEG data.

Our intra-participant (N = 17) results found no significant difference in P300 amplitude between conditions. However, after performing a median split based on the reaction time difference in high stress (PH)—low stress (GG) condition, we found two groups (Group 1 N = 9 and Group 2 N = 8) with contrasting performance and ERP results. Group 1, when exposed to heights, exhibited decreased target stimuli reaction time and P3 amplitude. Conversely, Group 2 exhibited increased reactionary performance with no change in P3 amplitude. Furthermore, we found a significant variation across stress levels in the Event-Related Spectral Perturbation (ERSP) for Group 1, while Group 2 remained consistent across different stress levels. Based on

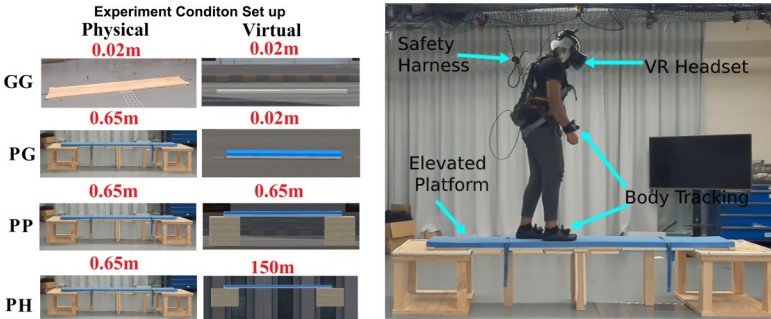

**Fig 1. The Experimental set up (both physical and virtual) and the equipment used for each condition with the respective elevation written in red.**

## A - Experiment Condition Timeline

**Fig 2.** (A) The timeline for each condition and the trials per condition. (B) An outline of a single walking trial (the laboratory pictures were taken before pandemic restrictions). The ground and elevated walking platform have the same walking layout as illustrated. (C) An outline of a single Oddball trial with the respective stimuli time periods at the bottom.

our findings and our understanding of the YD law, we conclude that the effect of stress on the P300 response may be related to the participants' perceived effort in the task. Participants who performed worse on the task had a lower amplitude P300 response when stressed. Therefore, the reactionary performance may indicate whether stress will decrease a person's P300 amplitude.

## Methodology

### Experimental design

**Physical space.** The main novelty of our experimental design is the elevated physical platform. The platform provided physical height and haptic feedback when walking. This physical platform has improved the participant's immersion and can reliably elicit a stress response [36, 38]. We set two requirements when designing this platform. The participant's safety was the first requirement, and the second was to introduce a sense of fear from the awareness of physical elevation. To ensure participant safety, we lined the corners and edges with protective foam and installed a harnessed fall arrest system (to prevent fall-related injuries). The platform's height (0.65 m) was set to the maximum allowable height based on the height of the rail, safety line, and harness system. We progressively designed the platform and rated the design to support up to 150 kg (experimental exclusion criteria restricted this to 95 kg); this was set through the failure of a previous iteration to support the specific weight.

The second concern was creating a sense of danger/fear on the platform. We introduced a small amount of instability through metal plates on the bottom of the platform. A surface foam

layer was introduced to add further postural instability when walking on the elevated platform. We found that instability factors added difficulty to walking, which increased the participant's anxiety levels. The ground platform consisted of a separate board with matched dimensions to the elevated platform. This correspondence ensured consistent gait behaviour during the experiment. The elevated (0.65 m height) and ground (0.02 m height) platforms had the exact walking space dimensions of 2.4 m long and 0.3 m wide.

**Virtual space.** We designed the Virtual Environment (VE) to correspond strongly with the experimental setup. The VE was developed using the Unity game engine. The physical plank's dimensions, orientation, and position were measured through the Optitrack motion capture (mocap) system (12 flex13 cameras) and mapped in virtual space. The usage of mocap ensured the accurate translation between the physical and virtual space. A calibration process was developed using the HTC Vive controllers, which capture any offset movement and vibrations of the physical platform and adjust the VE accordingly.

We chose to set the VE in an urban environment because the buildings and the overall city provided the participants with a believable environment for the virtual height. The buildings emphasised the sense of height by scaling the surrounding buildings. The high-quality VE was rendered through the HTC Vive Pro VR HMD, and a high-power GPU with sufficient processing was used (NVIDIA GTX1080 GPU, Core i7). A wireless adapter was added to improve the safety and mobility of the participants.

The participants used a virtual avatar as the medium for virtual embodiment. The avatar used inverse kinematics (FinalIK by RootMotion [39]) and a six-point body tracking system (one HMD, one waist tracker, two hand trackers, and two feet trackers) using HTC Vive trackers. The avatar improved the participant's spatial awareness of the VE and their sense of presence. Ambient urban environment noise was played in the background as the auditory stimuli.

## Experiment protocol

**Walking trials.** This experiment tested four conditions; each consisted of a combination of the physical (ground and elevated platform) and virtual (ground, elevated platform, and extreme height) independent variables. Fig 1 outlines each condition for this experiment; the conditions are *GG*, *PG*, *PP*, and *PH*. Every participant experienced the four conditions in a randomised sequence (Fig 2A). Due to timing constraints of the physical setup, the physical ground platform (*GG*) was randomised separately from the elevated platform (*PG*, *PP*, and *PH*). Each experimental condition consisted of 5 trials (1 baseline walk and 4 walking trials with 25 Oddball trials tasks). One walking trial constituted the trip from the starting position to the end and the return walks back to the start (see Fig 2B). Participants were instructed to walk in their natural gait. During the return trip for a non-baseline walk, the participant would perform 25 oddball trials in the middle of the platform.

**Oddball paradigm.** We selected visual oddball as the P300 task due to the reliability and simplicity of implementation. The oddball task consisted of a serialized sequence of images consisting of green (non-target) and blue (target) circles at a 1:4 target to the non-target ratio (see Fig 2C). The participant will complete 550 trials of the oddball task (3x50 trials baseline and 4x4x25 trials during each condition). All oddball trials were completed in a stationary position. The participants reacted to the target stimuli through a key press on the remote controller. All events were synchronized through a lab streaming layer (LSL) server. The same standing oddball task was performed after the experiment, along with a post-experiment questionnaire. The participants were given 3 minutes (minimum) resting periods between each condition; the participants may extend the rest time based on need. See Fig 2A for the timeline of the experiment.

The interval timings for the Oddball stimuli were decided based on previous visual oddball and rapid serial visual presentation p300 paradigms. The Sellers et al. [40] study assessed P300 performance at various intervals and target:non-target scenarios. The study found optimal P300 performance occurs when adequately long (interval > 350ms) stimuli reveal/hidden intervals and easily discernible images. We explored various ISI from previous works [41–44]. We selected the interval of 800ms stimuli to reveal and 400ms hidden with a block interval of 1000ms. We chose colour variation as an easily discernible target/non-target difference. Before the experimental conditions, the participants were trained on 50 training trials of the oddball task while seated (on the computer monitor) and standing (in VR).

During the conditions, the participants were continuously encouraged to wear the VR HMD throughout the experiment. This provided a continual sense of presence in the VE. The participants would step onto the physical platform in a VR calibration area and then be shown the appropriate scene for the condition. The participant would also be clipped onto the safety harness during the elevated walking conditions. During rest breaks, participants were encouraged to close their eyes to prevent eye strain and provided the option to remove the VR HMD if they felt severe discomfort. During the experiment, we maintained social distancing protocols and sanitised all equipment.

## Measurements

**Self-Assessment Manikin.**   We used a Self-Assessment Manikin (SAM) questionnaire to assess the arousal level of the participant after each walking trial. The valance and dominance scores were not collected due to the timing constraints of the experiment. We only collected arousal ratings based on pilot testing to minimise the time between walking trials. We selected the arousal component based on the close relationship between cognitive activity (affective arousal) and threat perception [16]. At the end of each trial, the participants would be verbally asked to rate their current arousal level based on the SAM figures, which score from 1–9 [45]. The participants were shown the SAM questionnaire figures before the experimental conditions. The SAM analysis involved separating the data by condition and averaging. The arousal level provides a reliable indicator of whether or not the participant has perceive a threat and provides an indication of stress from the threat perception [37, 46].

**Reaction time.**   The participant's reaction times to the stimuli were collected through the key-press remote controller. This key-press event was collected through an LSL key input stream that would synchronise the key-press events to the EEG data. During the data analysis, the RTs were calculated by the difference in latency between the stimuli appearing and the participant's key-press event. We filtered out trials with incorrect reaction events (multiple or non-target images) and RT outside of the reasonable range (300ms<RT<1200ms) [47]. Additionally, we correlated the RT and SAM results to explore potential trends for how stress affects RT performance.

## Electroencephalograph

**Apparatus.**   The participant's scalp voltage was recorded through a 64-channel (additionally, with 1 FCz reference and 1 ground electrode) actiCAP active electrodes cap encoded by the Liveamp EEG system. The Liveamp system was chosen due to the amplifier's portability, as it is a wireless system that can support high-density channel arrays. All electrode channels were gelled to have an impedance below 10 kΩ (recommended impedance for active electrodes is 25 kΩ and traditional passive electrodes are 5kΩ). The EEG data were collected at a 500 Hz sampling rate with all 64 channels active, and events were synchronised through LSL.

**Preprocessing.** The EEG data preprocessing aimed to clean line noise, muscle movement, eye blinking, and other walking-based artifacts. Our preprocessing pipeline was based on Mokoto's pipeline [48], Singh et al. [49], and Do et al. [50]. The data were processed on Matlab using functions based on the EEGLab toolbox by the Swartz Center for Computational Neuroscience [51]. The walking artefacts were reduced through Artifact Subspace Reconstructions (ASR) [52]. We re-referenced the data to the common average across the electrode channels. To avoid a rank deficiency problem, we re-added the FCz reference electrode. The ASR function reconstructed high noise portions of the EEG data with amplitudes of the 'cleanest' or lower noise portions. We then used the Adaptive Mixture Independent Component Analysis (AMICA) algorithm to compute the components within the EEG data and remove non-brain components such as eye blink and muscle artifacts. The data with the removed component is back-projected and analysed at a channel-based level. The EEG data was filtered to remove the head movement, walking, and complex stimuli artefacts.

The pipeline contained the following steps:

1. Bandpass finite-impulse response (FIR) filter 1–50Hz for Anti-Aliasing and removing high and low-frequency noise,

2. Down-Sample data to 250Hz to reduce computational time,

3. Clean line noise,

4. Apply ASR,

5. Replace removed channels by interpolating from neighbouring channels,

6. Add current reference back to the data and Re-reference data to signal common average,

7. Compute ICA weights and Spheres through AMICA,

8. Dipole fit components,

9. Classify components (IC labeling), and

10. Remove any IC components where the brain classification is not dominant, residual variance $>0.15$, and dipoles that are outside the brain model.

**P300 event-related potential and Event-Related Spectral Perturbation feature extraction.** The EEG data were epoched around the target stimuli event. The epoch was extracted 800ms (1200ms for the ERSP results) after the onset of the target stimuli with a baseline period of -400—0ms. These periods were determined by the stimuli display time (800ms) and the in-between stimuli time (-400ms) as shown in Fig 2C. Previous literature [17, 22, 53] indicated that the midline frontal to parietal electrodes were appropriate with a P300 detection window of 300–500ms. Based on grand average topography (parietal activation shown in topography results, Fig 7), ERP visual inspection, and prior literature (Table 1), we selected the Pz electrode for analysis of the P300 peak activity (compared to other midline electrodes used for P300 response, Cz and Fz).

We assess three critical EEG data metrics to determine the effect of stress on cognitive performance. Firstly, the EEG topography of activation. The topography provides insights into whether the P300 wave is correctly stimulated. We assessed the topography at 300—500ms (the typical onset period for a P300 peak [54]) and compared the average across each group. The P300 response is typically characterised by activation in the parietal region on a scalp topography.

The second metric is the ERP response. This result is a plot of the electrode amplitude over the time domain. To extract the ERP response, collated the epoched EEG data and eliminated any outlier, missed, or incorrect trial. We removed trials where the participant failed to react or the RT was outside the reasonable range (300ms<RT<1200ms, 1200ms is the ISI). The P300 signal peaks were detected by finding the ERP's local maxima between the typical P300 peak period of 300—500ms [54]. We removed any trials with the P3 peak occurring after the participant's RT response. After outlier/mistrial removal, we averaged the trials for each condition and compared them across conditions. This results in a total trial count of 5917 (out of 6800) or 348.06±49.13 trials (out of 400) per participant. The distribution of trials between the two groups was: 331.44±61.03 (out of 400) per participant for Group 1 and 366.75±22.49 (out of 400) per participant for Group 2.

As highlighted in Table 1, the previous research has mixed results on stress's effect on P300 behaviour. The ERP comparison allows the observation of the N170, P2, N2, and P300 peaks. The P300 peaks were correlated with the SAM ratings to observe if there is a direct relationship between stress and P300 amplitude.

The last metric is the Event-Related Spectral Perturbation (ERSP). The ERSP response measures change in frequency spectral behaviour over time. The spectral behaviour can provide insights into potential reasoning for behavioural change between conditions. The brain has five main frequency oscillation types that we can observe (after preprocessing filters, the frequency range is 1–50 Hz), Delta (1–4 Hz), Theta (4–8 Hz), Alpha (8–12Hz), Beta (13–30Hz), and low Gamma (30–50 Hz) [55]. The typical P300 ERSP response (from previous literature [56]) is characterised by the alpha and theta synchronisation during the onset (before) of the P300 peak and a gradual alpha and beta desynchronisation during the P300 peak. We extracted the ERSP response through the newtimef function provided by EEGLab. We selected a larger epoch window (-400 to 1200ms) to observe better the spectral behaviour after the onset of the P300 peak. The data window was set to 300 frames with the baseline set to the period -400 to 0ms. The time-frequency decomposition was calculated using Fast Fourier Transform (FFT) and Hanning window tapering. To compare the ERSP responses, we used the comparison function within newtimef to compute the difference between the conditions.

## Participants

The UTS human research ethics committee approved this human research experiment. The application ETH18–3098 was reviewed and approved by the committee. All participants provided written informed consent and received 60 Australian dollars in compensation for their participation (regardless of outcome or termination). We recruited 20 adults (5 women and 15 men) between the ages of 21 to 35. The mean age was 26, and the variance was 4.70. The key exclusion criteria were:

- an inability to understand the experimental instructions (language and cognitive ability),

- existing medical conditions, such as

  - neurological and cardiovascular disorder,

  - diagnosed mental health issues (depression, anxiety, or chronic stress),

  - sensory (visual, vestibular, or auditory) dysfunction, and

  - gait (unable to perform unassisted walking) disorders.

- weighs more than 95 kg (participant safety).

The data analysis only included a dataset of 17 participants. One participant's (man) data were excluded due to incomplete data from hardware failure. Another participant (man) felt overwhelmed by the height exposure and did not complete the *PH* condition. One more participant (man) was excluded due to failure to complete the oddball task for one condition.

## Dividing participants into groups

Fig 3 shows the results for the difference in the RT average between the conditions *GG* and *PH* (*PH-GG*) for each participant. Based on the results, we observe two distinctive outcomes when the participants are in a stressful environment. Upon further examination of the participant's behaviour and previous works relating to changes in RT [55], it became clear that certain individualised factors must contribute to the variance in behaviours when experiencing stress. To further explore the contrasting RT behaviour, we performed median split [57, 58] on the sample based on the difference in RT in the conditions *GG* and *PH* (as seen in Fig 3). As outlined by Iacobucci et al. [58] and other EEG studies [59], the use of the artificial categorisation through the median split was to separate the distinctively different participants (slower RT when stressed vs faster RT when stressed). Group 1 consisted of the participants who performed worse (slower RT) when exposed to the stressor, while Group 2 showed improvement/unaffected RT performance regardless of the condition. This participant grouping was used for the EEG-related datasets. Table 2 presents the demographic details of the two groups. A limitation of this approach is the confounding factors that may be unrelated to stress, such as the participant's engagement in the task, fatigue levels, and learning effect. While all participants did successfully complete the study with moderate to good target identification accuracy (Group 1 with 331.44±61.03 successful trials and Group 2 with 366.75±22.49 successful trials, 400 total trials), the inclusion of more comprehensive measurements for fatigue and user engagement can better reducing the confounding factors of this method.

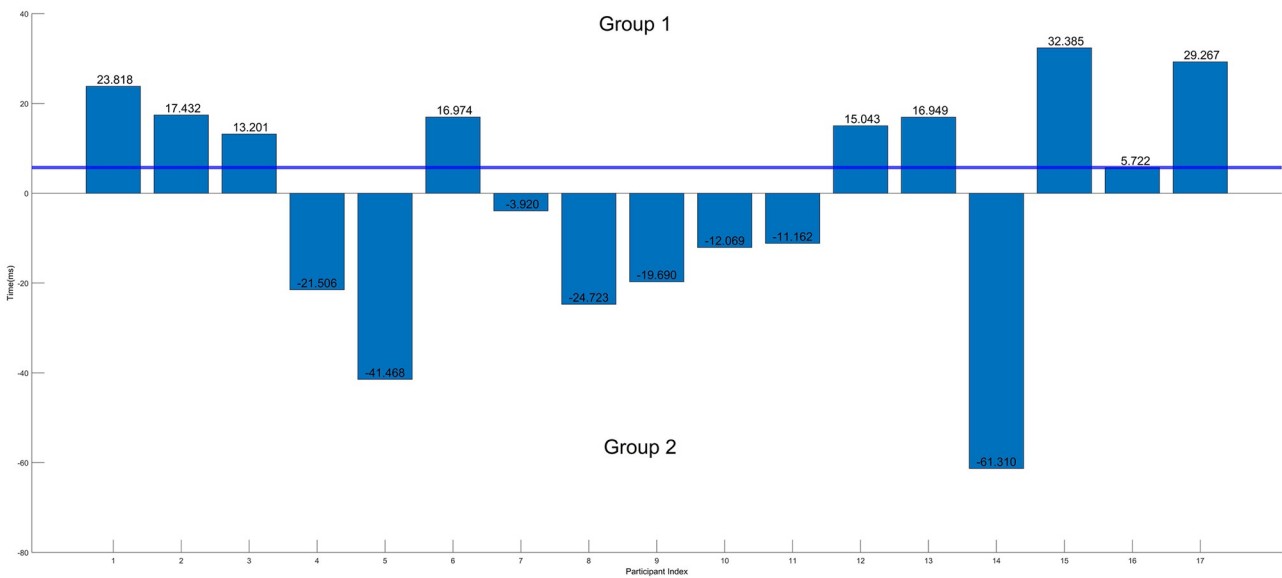

**Fig 3. This figure shows the average RT difference between the conditions *GG* and *PH*(*PH-GG*) for each participant.** The horizontal blue line represents the median value. The two groups categorized in this study can be observed, with Group 1 being the participants with a slower RT when elevated (positive difference) and Group 2 with a faster RT (negative difference).

Table 2. The two groups from the Reaction Time (RT) and their demographic information.

| Group | N | Sex | Mean Age | Variance Age |
|---|---|---|---|---|
| 1 | 9 | 6 men 3 women | 25.00 | 4.64 |
| 2 | 8 | 6 men 2 women | 26.63 | 5.15 |

## Statistical analysis

The normality of the data was determined by a one-sample Kolmogorov-Smirnov test (compared against the cumulative distribution function). The RT, SAM, and frequency power band data were not normally distributed. For these metrics, we applied a Friedman's test with a post hoc Tukey-Kramer HSD (honestly significant difference) test for pairwise comparison. We used a two-tailed Wilcoxon rank-sum test for the between-group analysis. The RT, P300, and SAM correlation R and P-value were calculated using the Spearman test since the data was non-parametric and monotonic. The EEG topography data was calculated through a one-way ANOVA with the Tukey-Kramer HSD test for each channel between conditions.

The statistical results for the ERP were calculated using Friedman's test with the post hoc Tukey-Kramer HSD (honestly significant difference) test, which compared each data point between the participants and conditions [60]. The significance criteria were set at $\alpha < 0.05$, and we only considered periods larger than 20ms as significant. For the ERSP response significance mask was calculated through bootstrap permutation ($\alpha < 0.05$) when compared to the baseline; the p values were false discovery rate (FDR) corrected. The significance stars for the figures are *p<0.05, **p<0.01, and ***p<0.001.

## Results

### Self Assessment Manikin

Fig 4 shows a plot of the SAM results for all participants, Group 1 and Group 2. The SAM response of all participants (*GG* M = 1.33 and SD = 0.92, *PG* M = 2.31 and SD = 1.60, *PP* M = 2.53 and SD = 1.53, and *PH* M = 4.95 and SD = 2.25) found significant difference between the four conditions ($\chi^2(3,67) = 37.88$, p<0.001). The post hoc Tukey test found a significant difference when comparing the *GG* condition to the *PP* condition (p = 0.047) and the *PH* condition (p<0.001). We also found a significant difference between *PH* compared to the *PG* (p = 0.002) and *PP* (p = 0.002). No other significant differences were found (p>0.057).

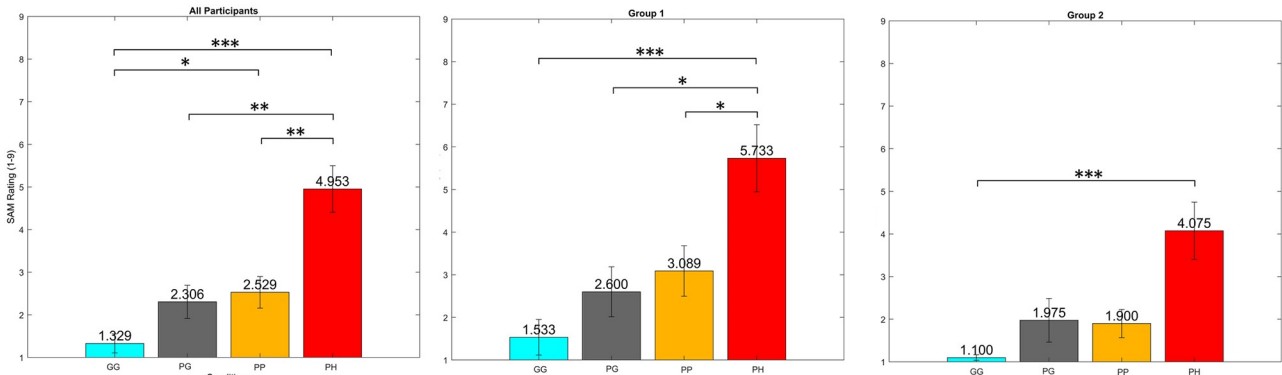

Fig 4. The average with standard error bars for the SAM responses for each condition across all participants, Group 1, and Group 2. Significance Star P-values *p<0.05, **p<0.01, and ***p<0.001.

The SAM response for Group 1 (*GG* M = 1.53 and SD = 1.25, *PG* M = 2.60 and SD = 1.76, *PP* M = 3.09 and SD = 1.78, and *PH* M = 5.73 and SD = 2.36), we found a significant difference between the four conditions ($\chi^2$(3,35) = 22.41, p<0.001). The post hoc Tukey test found a significant difference between the *PH* condition and the other conditions *GG* (p<0.001), *PG* (p<0.05), and *PP* (p = 0.04). No other significant differences were found (p>0.148).

The SAM response for Group 2 (*GG* M = 1.10 and SD = 0.19, *PG* M = 1.98 and SD = 1.45, *PP* M = 1.90 and SD = 0.94, and *PH* M = 4.08 and SD = 1.90), we found a significant difference between the four conditions ($\chi^2$(3,31) = 15.61, p = 0.001). The post hoc Tukey test found a significant difference between the *PH* condition and the *GG* conditions (p<0.001). No other significant differences were found (p>0.062).

When comparing the SAM rating between groups, we found no significant difference between the groups for every condition (W>98.50, Z<1.64 and p>0.102).

## Reaction time

Fig 5 presents a violin plot of the RT results of all participants, Group 1, and Group 2. The RT for all participants (*GG* M = 497.40ms SD = 48.09, *PG* M = 499.82ms SD = 54.12, *PP* M = 494.16ms SD = 46.19, and *PH* M = 495.93ms SD = 39.94) found no significant difference between the four conditions ($\chi^2$(3,67) = 1.02, p = 0.796). The post hoc Tukey test found no significant difference when comparing the four conditions (p>0.789).

The RT for Group 1 (*GG* M = 474.24ms and SD = 48.92, *PG* M = 492.71ms and SD = 67.68, *PP* M = 485.95ms and SD = 57.42, and *PH* M = 493.22ms and SD = 49.11), we found a significant difference between the four conditions ($\chi^2$(3,35) = 11.53, p = 0.009). The post hoc Tukey test found a significant difference between the *PH* condition and the *GG* condition (p = 0.006). No other significant differences were found (p>0.082).

The RT for Group 2 (*GG* M = 523.45ms and SD = 32.93, *PG* M = 507.82ms and SD = 36.34, *PP* M = 503.40ms and SD = 30.40, and *PH* M = 498.97ms and SD = 29.50), we found a significant difference between the four conditions ($\chi^2$(3,31) = 8.55, p = 0.036). The post hoc Tukey test found no significant difference between each condition (p>0.057).

When comparing the RT results between groups, we found a significant difference between Group 1 and Group 2 for the *GG* (W = 60, Z = -1.97, p = 0.048) conditions. No difference was found in the other conditions (W>69.00, Z<-1.11 and p>0.269).

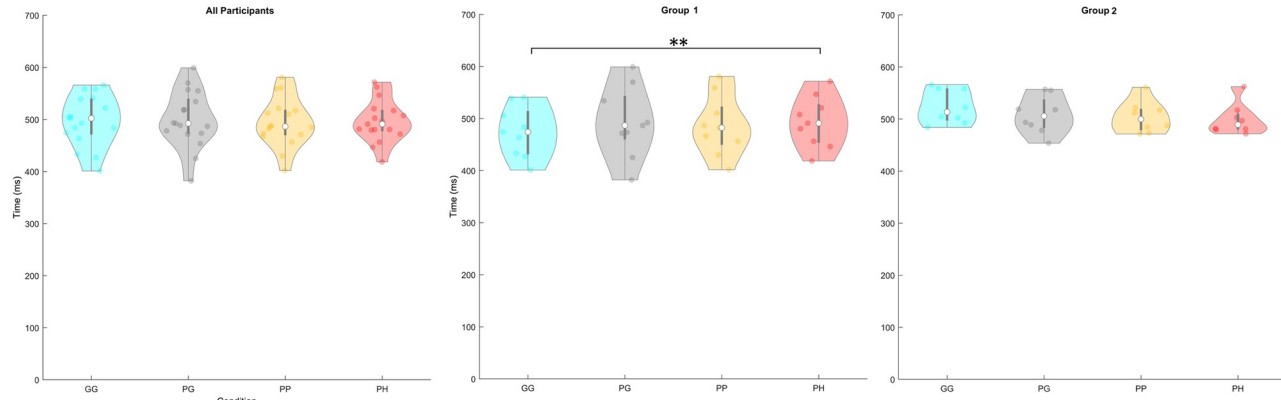

**Fig 5. Participant's average target image RT (0ms = start of visual stimuli) for each condition across all participants, Group 1, and Group 2.** Significance Star P-values *p<0.05, **p<0.01, and ***p<0.001.

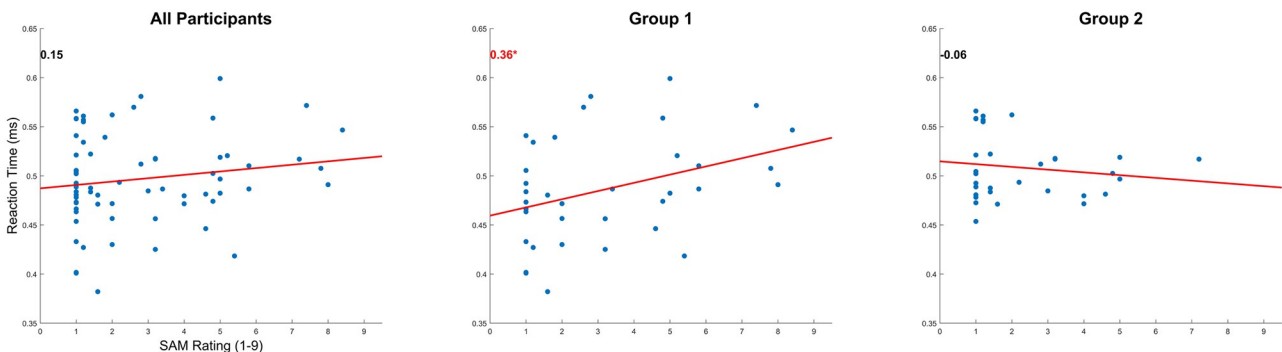

**Fig 6. Participant's RT and SAM correlation plot by the average value of each subject.** R Value is on the top left of each figure (Spearman test). Significance Star P-values *p<0.05, **p<0.01, and ***p<0.001.

## Correlation SAM-RT

Fig 6 illustrates the correlation between the SAM rating (arousal/stress) and RT for all participants, Group 1, and Group 2. No significant correlation was found in all participants (R = 0.15, P = 0.2378) and Group 2 (R = -0.06, P = 0.761). A significant correlation was found in Group 1 (R = 0.36, P = 0.030). The trend suggests that the RT will also increase as SAM increases, reflecting decreased performance.

## Topography and Event-Related Potential (ERP)

Fig 7 depicts the topography of each condition between 300ms to 500ms for Group 1 and 2. The topography confirms that the P300 task activates the brain's parietal region. We did not

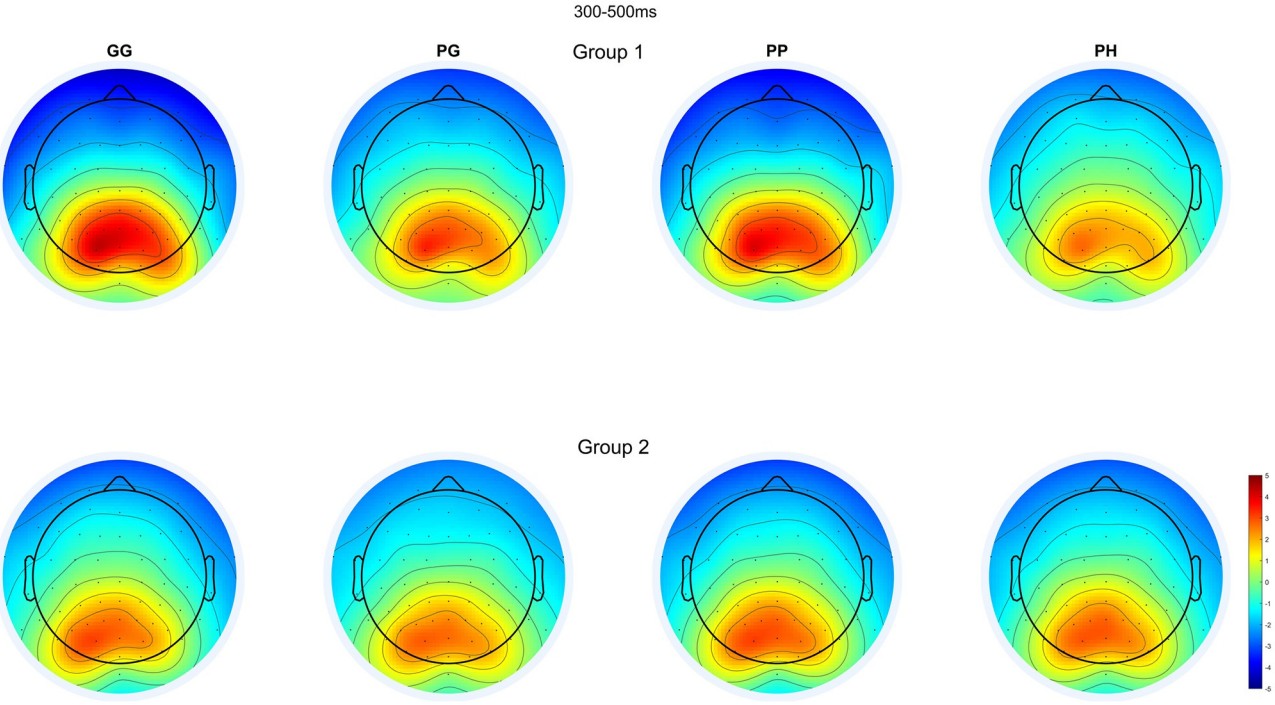

**Fig 7. The grand average EEG Scalp Map topography for each condition averaged between 300ms to 500ms (0ms at the start of target stimuli).**

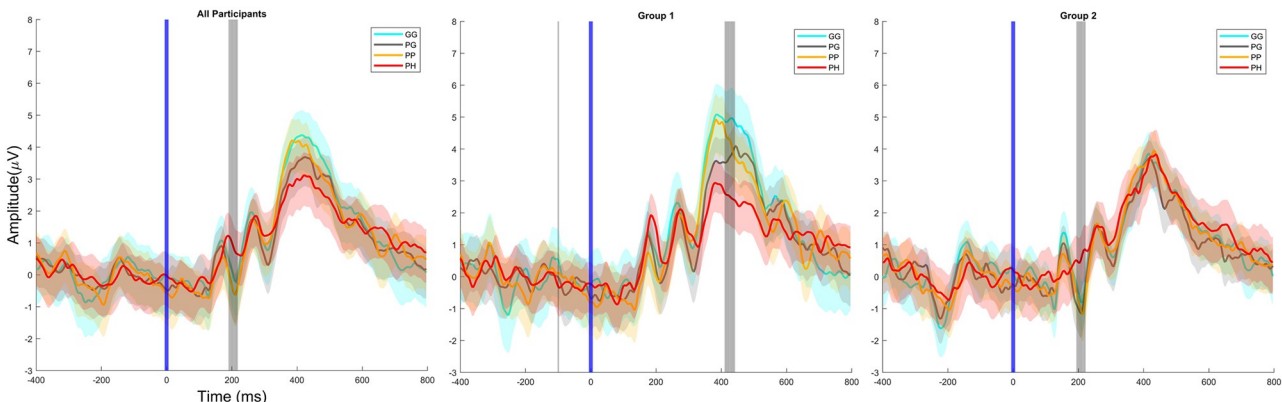

**Fig 8. The grand average Pz channel ERP response epoched (-400ms to 800ms) around the target stimuli (0ms) for each condition across all participants, Group 1, and Group 2.** The color bars indicate the confidence interval of 95%, and the gray bars show the significant difference between at least two conditions.

observe any significant difference between conditions, suggesting that the region of activation is not significantly changed between different experimental conditions (F(3,35)<1.58, p>0.213).

Fig 8 presents the ERP response from the Pz channel. When comparing the ERP response for all the participants, we observed a significance around the N170 peak during the 192ms—216ms period. We found a significant difference between the four conditions ($\chi^2$(3,67) = 14.44, p = 0.002). The post hoc Tukey test found a significant increase in N170 amplitude for the *PH* condition compared to *PG* (p = 0.005) and *PP* (p = 0.040). No other significant differences were found (p>0.310).

For the ERP responses of Group 1, we found a significance around the P300 peak during the 412ms—456ms period. We found a significant difference between the four conditions ($\chi^2$(3,35) = 11.40, p = 0.001). The post hoc Tukey test found a significant decrease in P300 amplitude for the *PH* condition when compared to the *GG* condition (p = 0.006). We also observed a decrease in during *PG* condition when compared to the *GG* condition (p = 0.030). No other significant differences were found (p>0.126).

For the ERP responses of Group 2, we found a significance around the N170 peak during the 196ms—220ms period. We found a significant difference between the four conditions ($\chi^2$(3,31) = 13.20, p = 0.004). The post hoc Tukey test found a significant increase in N170 amplitude for the *PH* condition compared to *PG* (p = 0.006) and *PP* (p = 0.011). No other significant differences were found (p>0.144).

## Correlation P300 peak amplitude—SAM

Fig 9 illustrates the correlation between the SAM rating (arousal/stress) and P300 peak amplitude for all participants, Group 1, and Group 2. We found a significant correlation when comparing all the participants (R = -0.24, P = 0.046). We did not observe any significant correlation when splitting the data into Group 1 (R = -0.27, P = 0.113) and Group 2 (R = -0.31, P = 0.080).

## Event-Related Spectral Perturbation (ERSP)

Fig 10 shows the Pz channel ERSP response between Group 1 and Group 2. Fig 11 illustrates the comparison between conditions for both participant groups. In Group 1, during the P300

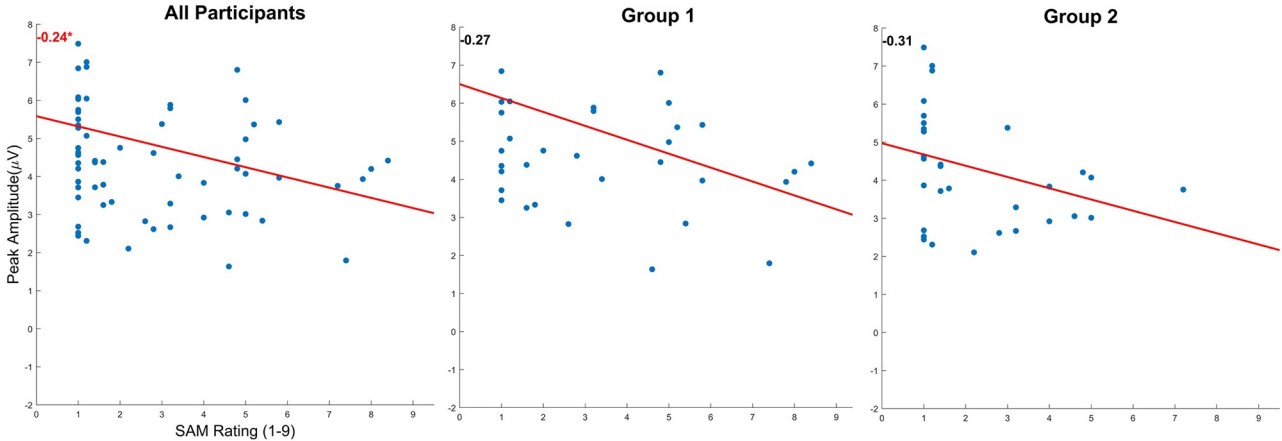

**Fig 9. Participant's P300 peak amplitude and SAM correlation plot by the average value of each subject.** The R-Value is on the top left of each figure (Spearman test). Significance Star P-values *p<0.05, **p<0.01, and ***p<0.001.

period (300ms—500ms), we can observe a desynchronisation of the delta and theta band. When comparing *PH* to *GG* and *PP* during (300ms—500ms) and after (500ms—1000ms) the P300 period, we observe a synchronisation of the beta and gamma bands. Interestingly, the beta and gamma activity is not observed when comparing *PH* to *PG*. For Group 2, we do not observe the synchronisation of the beta and gamma bands. After the P300 period (500ms—1000ms), an alpha and mu (8–13 Hz [61]) synchronisation can be observed when comparing *PH* to the other conditions (*GG*, *PG*, and *PP*).

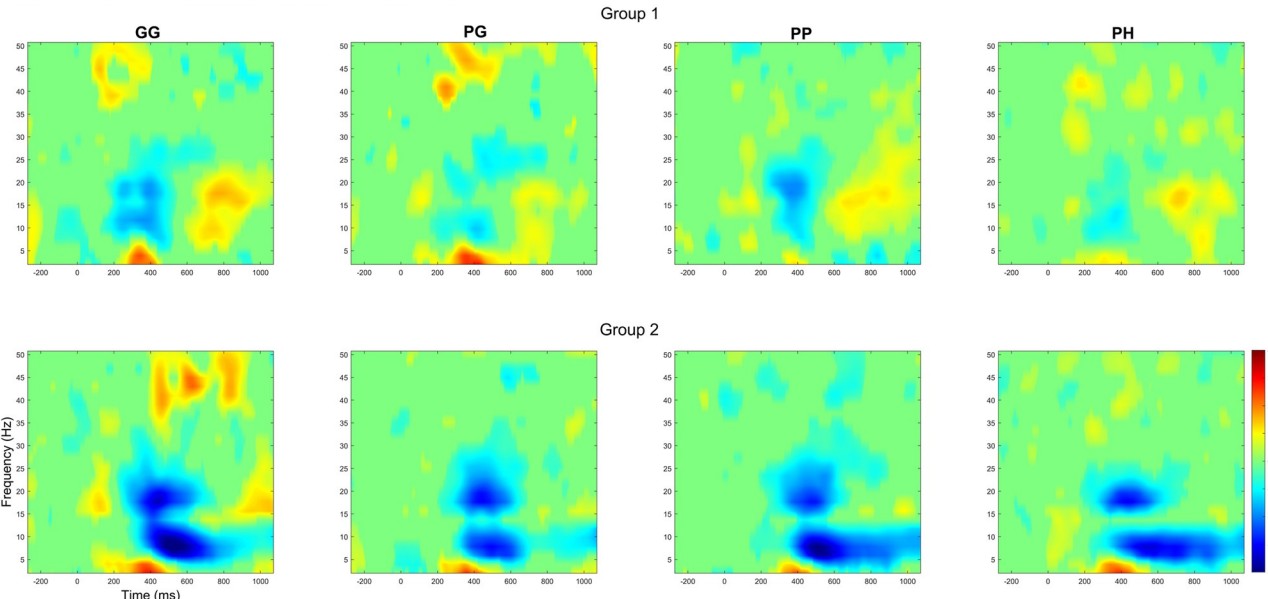

**Fig 10. The grand average ERSP response for the PZ channel for each group and condition.** The ERSP plots are masked using a significance mask with the criteria of $\alpha < 0.05$ and FDR correction.

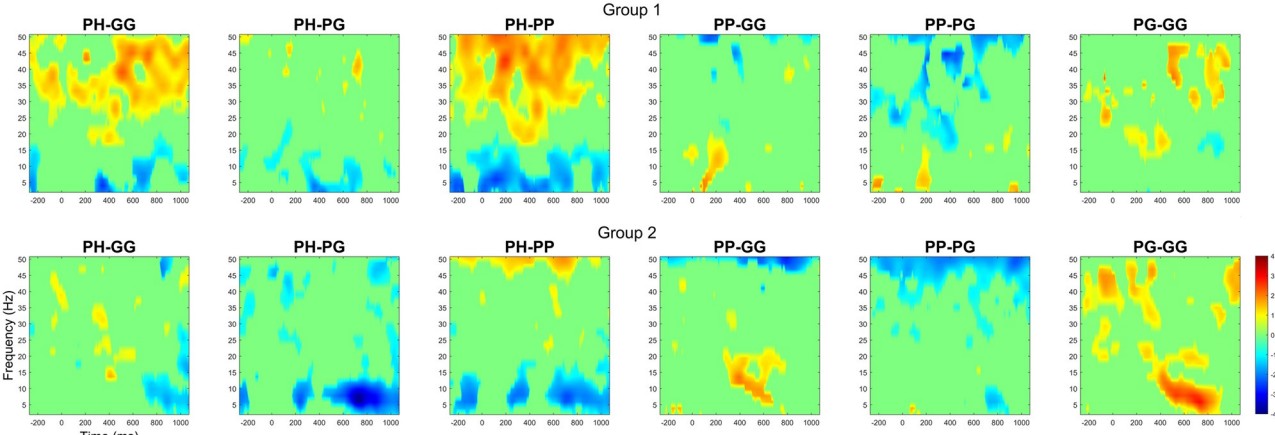

**Fig 11. Analysis of ERSP response for the PZ channel by comparing each condition in a pairwise manner.** Each ERSP represents the difference between two conditions (condition 1—condition 2). The ERSP plots are masked using a significance mask with the $\alpha < 0.05$ criteria.

## Discussion

### Validity of stress and P300 response

The efficacy of the stress elicitation paradigm is a key component in ensuring the validity of the effects of stress on cognitive performance [4]. The SAM results for All Participants (Group 1 + Group 2) and Group 1 and Group 2 separately (Fig 4) indicate a significant increase in rating as the virtual and physical height increases. This result indicates that the *PH* condition is the most stressful, *PP* and *PG* condition at medium levels, and *GG* is the least stressful.

The second factor is the reliability of the oddball paradigm in producing a P300 wave response. The P300 wave is distinct in its parietal region activation and distinct ERP wave peak [62]. The EEG scalp topography (Fig 7) corroborates this result as a parietal activation can be observed in all the experiment conditions and baselines. The P300 waveform can also be observed in the ERP responses in Fig 8. The waveform conforms to the traditional shape with clearly identifiable N170, P2, N2, and P300 peaks [54] within all the conditions.

### Relation between stress and P300

No significance could be found when observing the RT and P300 ERP responses on the sample group level (all Participants). However, we find two starkly different behaviours if we observe the data from the group-level perspective. As outlined in the SAM results, both Group 1 and Group 2 experience a stress response during the *PH* condition (when compared to *GG*). Neither group rated the conditions differently when comparing Group 1 and Group 2, suggesting that both groups experience similar subjective stress responses. Both groups also demonstrate a P300 wave from the ERP and topography. The only inter-group difference is the RT performance for the condition *GG*. This would suggest that Group 1 has a faster RT than Group 2 when not under stress. This RT would then decline as the height increases.

Interestingly, during the stressful condition (*PH*), Group 1's performance significantly deteriorates. Conversely, Group 2's performance improves when in walking at heights. The correlation of SAM to the RT data provides a clue into the relationship between the experienced stress response and RT performance for each Group. Group 1 found a strong correlation (see Fig 6) between the SAM and RT data. Based on the linear fitting, it can be observed that the participants would gradually perform worse as the stress level increased. Group 2 did not

demonstrate any correlation between SAM and RT, which suggests that their reactionary performance was less affected by stress.

The ERP response and P300 peak data corroborate the findings of the RT data for both Group 1 and Group 2. It is evident that Group 1's P300 peak amplitude significantly decreases when experiencing the stress response. This outcome signifies that stress indeed decreases the amplitude of the P300 peak, which affirms the theory that stress decreases P300 amplitude [20]. However, Group 2 displays a distinct consistency level in ERP and P300 amplitude for the oddball task, which contends stress does not affect the P300 amplitude [27]. Incidentally, either Group's results correlate with past literature, as previously emphasised in Table 1.

Previous studies such as Jiang et al. [3] and Mingming et al. [24] reasoned that Group 1's ERP behaviour is due to the stress response acting as a distracter which reduces the participant's attention resources, engaged. This explanation is consistent with established work by Gray et al. [63] and Giraudet et al. [64] that affirms the relation between P300 amplitude and attention. On the other hand, this rationale does not explain the seemingly contradicting results of Group 2, which correlated with other studies [25–27]. The studies that support Group 2 rationalise that stress increases an individual's alertness, improving performance and increasing or maintaining consistent P300 peak amplitude.

Both of these explanations are further upheld by the ERSP results (Fig 10). Group 2 exhibited a remarkably consistent spectral behaviour that is typical of normal P300 behaviour. This consistency would suggest an undisrupted mental state [27]. The same consistency is not observed in Group 1. The ERSP response illustrates a change in spectral activity with the Beta power significantly increasing (Pz) during the *PH* condition. Based on prior research, an increase in Beta power may indicate a change in mental activity and resource allocation [65]. Schmidt et al. [66] propose that frontal-central beta is closely linked to memory resource allocation and thought processing. This finding may indicate a change in the neurological pathway for target recognition when experiencing stressors, in which Group 1 would rely more heavily on memory and mental resources than Group 2.

In summary, we found two groups exhibited seemingly contradictory P300 behaviours when experiencing height exposure. An elementary approach would be to dismiss the results of a group and support the other. However, the validity of past works and results from this experiment does not provide sufficient evidence to dismiss either behaviour. Both behaviours exhibited could be better explained by exploring the factors of contrast between the two groups. Factors, including fatigue, workload, and participant task engagement levels, may have contributed to the difference in behaviour between the two groups.

## The effectiveness of stress elicitation method

In order to rationalize both behaviours, we must analyze and identify the potential circumstances in which either behaviour might occur. Kamp et al. [20] identified two factors that may influence the experience of stress and P300 behaviour. Kamp et al. [20] proposed one factor: the stressor's effectiveness in inducing adequate stress levels. The paper suggested that TSST may have been more effective at eliciting stress; thus, decreasing P300 amplitude is the more likely behaviour. However, in the case of the Dierolf et al. [25] study, TSST yields a contradictory result (increase in P300 amplitude). Hypothetically, there is a difference between Groups 1 and 2 due to the effectiveness of the stressor. In that case, there should be an observable and significant difference in the SAM response between conditions and groups. This was not found in Fig 4, since both Group 1 and Group 2 exhibited signs of stress. Nevertheless, there is a subjectivity to the experience of stress. Group 1 and Group 2 participants can vary in stress/arousal while in stressful conditions.

Another factor is the participant's demographics, precisely age and sex [20, 67, 68]. Both age and sex can significantly affect the perception of stress and P300 amplitude. Table 2 presents the age and sex distribution between the two groups. We found no notable difference in demographics. Therefore, the participant demographics are unlikely to explain the difference between Group 1 and Group 2.

## Understanding Yerkes-Dodson law and participant's performance during the experiment

We propose to explain the difference between Group 1 and Group 2 through concepts derived from the YD law. Researchers [69] found that the difficulty and complexity of a task affect the shape of the YD curve. Difficult tasks that require more cognitive resources will require less/ low stress to reach optimal performance and then gradually decrease as stress increases. Simpler or easier tasks that can be performed more autonomously tend to remain stable as stress increases gradually and require much higher/extreme stress stimuli to cause a drop in performance. Based on the YD law, we propose two hypothetical explanations.

One explanation is that Group 1 and Group 2 have a subjectively different experience of stress [70]. Then, Group 2 may have experienced a less adverse experience of the stress response; thus, they were at the near-optimal performance level and yet to suffer the performance decrease. This trend can be observed by the improved performance (faster RT) in the *PH* condition and the consistent ERP and ERSP response. Conversely, Group 1 may have experienced a negative stress experience, thus passing the optimum levels with increased stress, decreasing performance and P300 amplitude (similar to the correlation in Fig 6). This behaviour is similar to the 'eustress'(low-stress) and 'distress'(high-stress) groups proposed by Bak et al. [70]. Their work suggests that subjects experiencing distress can exhibit behavioural and neurological recession to task performance response, while the eustress may not exhibit change (or even better performance). This hypothetical is plausible, but this explanation is difficult to prove due to the lack of significant difference between SAM scores and valance metrics.

Another explanation could be that the stress has affected the participant's perception of the task's difficulty and mental resource allocation, thus causing the division between Group 1 and Group 2. A similar concept was explored by Sellers et al. [40], who found that the degree of difficulty (exerted effort) of the P300 task would affect the P300 amplitude (higher difficulty caused decreased amplitude). Based on the ERP and ERSP results, it seems that Group 1 has allocated more mental resources (Beta increase) and struggled to complete the task under stress (less attention resource shown by the decreased P300 amplitude [3, 24]).

In contrast, Group 2 seems unaffected by stress and did not exhibit a difference in RT and P300 peak amplitude behaviour. Therefore, the participant's ability to cope with stress when performing the P300 task can affect the outcome of the P300 response. One theory to the coping mechanism could be indicated through the increase in the N170 peak for the Group 2 participants during the *PH*. As suggested by Bauer et al. [71] and the findings of Kropotov's study [72], the N170 behaviour is closely related to personal memory for visual stimuli recognition (also tied closely to threat perception). The increase in N170 peak amplitude could suggest that the Group 2 participants, when exposed to stressors, are more cognitively active or remain focused (or even perform better under stress). In the case of Group 1, the change in brain dynamics signifies poorer/decreased cognitive performance and higher susceptibility to stress. Thus, they are more likely to decrease RT performance when exposed to stressors.

Suppose we assume all the participants remained well-engaged with the experiment and had similar fatigue levels. In that case, both explanations can reconcile the conflict in previous

works as we have found both results to be true in different subjects. The results better support the second explanation. This explanation suggests we can use RT performance and brain behaviour to gauge and estimate a participant's susceptibility to stress and the expected change in their cognitive performance. Our strategy differs from previous research in that it metricises and classifies individual behaviour types rather than suggests a population-based trend. We can observe how stress affects people individually by metricizing the beta powers and P300 peak behaviours.

## Limitations

### Group sample size

Overall, this experiment contains a low sample size and thus may not accurately reflect the population's behaviour. The target sample for this study has initially been forty participants. However, due to the length of the experiment (2–3 hours) and the restrictions during the COVID pandemic, we were only able to recruit twenty participants and further excluded three due to technical (hardware failure), behaviour (not correctly completing the oddball task), and early termination (the *PH* conditions being too stressful). The seventeen participants were split into two groups, reducing the sample size. Hence, we do not want to speculate on the population claims but primarily assess the difference between Group 1 and Group 2. In the future, we hope to collect a larger sample group with broader demographics to validate the results better.

### Unbalanced factors

Another fundamental limitation is the unbalanced size, condition order, and demographics of Groups 1 and 2. Ultimately, this split grouping was a data-driven discovery during analysis. The size, condition order, and demographics were not matched. Unfortunately, this group split could not have been anticipated as prior studies yielded a singular behaviour, and pilot testing did not show this divergent behaviour. The unbalanced nature may hide factors such as fatigue, habituation, and demographic effects from our data analysis. Future studies should consider classifying participant behaviour and ensuring counterbalancing of conditions.

## Conclusion

Our study's findings conclude that the participant's susceptibility to stress dictates the P300 behaviour when exposed to a stressor. This explanation of individual differences within the participant group could reconcile the division within the literature on the P300 behaviour during the stress response. When observing the RT and P300 amplitude results, we found two groups exhibiting contrasting behaviours that correlate to previous studies. By observing the difference between the two groups, we can reconcile the conflicting reports of previous works. While we did not find any difference between the participant's demographics, we did find a significant change in the beta and gamma power within Group 1 that does not occur in Group 2. We speculate that there is an individualized difference in the P300 response when in a stressful condition. When under stress, certain people may experience increased effort or difficulty during the P300 task. Future works should consider including measurements for factors such as participant engagement and fatigue to isolate the factors that may affect a person's P300 response. Additionally, measurements including RT performance, P300 peak amplitudes, and beta/gamma power can provide an indication of the emotional and cognitive state of the person when exposed to stressors.

## Supporting information

**S1 Questionnaire. The SAM questionnaire sheet that was used during the experiment.**
(PDF)

## Author Contributions

**Conceptualization:** Howe Yuan Zhu, Hsiang-Ting Chen, Chin-Teng Lin.

**Data curation:** Howe Yuan Zhu.

**Formal analysis:** Howe Yuan Zhu.

**Funding acquisition:** Chin-Teng Lin.

**Investigation:** Howe Yuan Zhu, Hsiang-Ting Chen, Chin-Teng Lin.

**Methodology:** Howe Yuan Zhu, Hsiang-Ting Chen.

**Project administration:** Howe Yuan Zhu.

**Supervision:** Hsiang-Ting Chen, Chin-Teng Lin.

**Visualization:** Howe Yuan Zhu.

**Writing – original draft:** Howe Yuan Zhu.

**Writing – review & editing:** Howe Yuan Zhu, Hsiang-Ting Chen, Chin-Teng Lin.

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
