## [Decision Letter · Decision Letter 0]

23 Aug 2022

PONE-D-22-20226Understanding the Effects of Physiological Stress on the P300 ResponsePLOS ONE

Dear Dr. Zhu,

Thank you for submitting your manuscript to PLOS ONE. After careful consideration, we feel that it has merit but does not fully meet PLOS ONE’s publication criteria as it currently stands. Therefore, we invite you to submit a revised version of the manuscript that addresses the points raised during the review process.

We look forward to receiving your revised manuscript.

Kind regards,

Humaira Nisar

Academic Editor

PLOS ONE

Journal Requirements:

2. Please change "female” or "male" to "woman” or "man" as appropriate, when used as a noun (see for instance https://apastyle.apa.org/style-grammar-guidelines/bias-free-language/gender).

"This work was supported in part by the Australian Research Council (ARC) under discovery grant DP210101093 and DP220100803. Research was also sponsored in part by the Australia Defence Innovation Hub under Contract No. P18-650825, US Office of Naval Research Global under Cooperative Agreement Number ONRG - NICOP - N62909-19-1-2058, and AFOSR – DST Australian Autonomy Initiative agreement ID10134. We also thank the NSW Defence Innovation Network and NSW State Government of Australia for financial support in part of this research through grant DINPP2019 S1-03/09 and PP21-22.03.02."

"This work was supported by Australian Research Council (ARC, https://www.arc.gov.au/) (DP210101093 and DP220100803), Australia Defence Innovation Hub (https://innovationhub.defence.gov.au/, P18-650825), US Office of Naval Research Global (https://www.nre.navy.mil/, ONRG - NICOP - N62909-19-1-2058), AFOSR – DST Australia Autonomy Initiative (ID10134), and NSW Defence Innovation Network (https://innovationhub.defence.gov.au/, DINPP2019 S1-03/09 and PP21-22.03.02) to Chin-Teng Lin. The funders had no role in study design, data collection and analysis, decision to publish, or preparation of the manuscript."

Reviewers' comments:

Reviewer's Responses to Questions

**Comments to the Author**

1. Is the manuscript technically sound, and do the data support the conclusions?

Reviewer #1: Partly

Reviewer #2: No

Reviewer #3: No

2. Has the statistical analysis been performed appropriately and rigorously? 

Reviewer #1: No

Reviewer #2: I Don't Know

Reviewer #3: No

3. Have the authors made all data underlying the findings in their manuscript fully available?

Reviewer #1: No

Reviewer #2: Yes

Reviewer #3: Yes

4. Is the manuscript presented in an intelligible fashion and written in standard English?

Reviewer #1: Yes

Reviewer #2: Yes

Reviewer #3: Yes

5. Review Comments to the Author

Reviewer #1: The experimental setup is interesting. However, this reviewer has some concerns which have to address during major revision.

1. The author should define "stress" in this context clearly. It would be better to replace "stress" by the word which can be measured quantitatively and/or having the standard questionnaire in identifying the level of mental or cognitive world load during interested periods. This famous related review article might help P. Consumer Grade EEG Measuring Sensors as Research Tools: A Review.

2. P300 or ERSP visualization and analysis need further improvement and exploration. The recent work with patients might help the author in the improvement of this point. This reviewer suggest the author to study more from A Pilot Study on Visually Stimulated Cognitive Tasks for EEG-Based Dementia Recognition.

3. Also the analysis of EEG to together with response or reaction times can learn from: Supervision of a self-driving vehicle unmasks latent sleepiness relative to manually controlled driving.

Looking forward to reading the revised version.

Reviewer #2: This is an interesting study on P300 responses under real and virtual height exposure. Basically, the experimental design is somewhat unbalanced (and hard to understand from the description) but it’s a good approach towards a quite timely question. And this is the main critique from my side. This is a nice pilot for a completely controlled study, but, in particular with some minor faults in the design and a sample size of 17 that additionally is divided into two groups, this is not sufficient for a publication. Along with this criticism, it should be noted that the main effect (the modulation of P3 amplitude in one of the two groups) just touches significance (Z=1.96, p=0.049) although the authors seem to have renounced to correct their data against alpha inflation.

As the authors claim in the final section, they were kind of surprised by the interindividual differences in behavior. Those are, besides the fact that mobile EEG is used in order to investigate more realistic behavior, an interesting point of this study, however, the authors nowhere provide evidence that there are really two distinct groups. They simply divided their sample by a split half on one RT parameter, which was for sure not bimodally distributed. The reader might have the impression that they searched for significance based on different analyses strategies and this one turned out to provide a p value below .05.

The entire MS appears to be intended in another context as presented. The introduction tlks a lot about BCIs, which are nowhere touched experimentally. This topic does not return before the final conclusion. The entire discussion section focuses on stress and cognition, which in my mind is the central issue here.

Minors

• The P300 paradigm is poorly described and not a standard for the P300 literature. Maybe this approach is usual in connection with PCIs, but the majority of visual oddball tasks uses much shorter stimulus presentations, longer interstimulus intervals which are normally additionally jittered.

• Why was the SAM presented before the trials? Would not a survey after performing the task better reflect stress?

• Removing any component that is less than 85% classification for brain component (done with ICLabel?), is a hard criterion. How many ICs remained in the data then?

• The authors appear to make separate tests for within group effecs and between groups effects on the same parameter. Was any correction against alpha error inflation applied?

• statistical parameters are missing in many places. Also not significant effects should be presented with their numbers.

• Another evidence for effect searching: In the introduction they speak about Alpha and Theta (“276 pected P300 ERSP response is the alpha and theta synchronisation during the onset (before) of the P300 peak and a gradual beta desynchronisation during the P300 peak”), in the result section they present an effect in Beta power, and in the conclusion they added gamma activity. When I look at the time frequency plots, I rather see Mu responses in Group 2, which might reflect encapsulation due to the balance task.

Reviewer #3: This study tries to address the influence of stress under a naturalistic environment on the P300 response. This research is very practical, and the paradigm design is also very interesting. However, many critical problems of statistical analysis in this manuscript make the results of this study unconvincing. Carefully checking the processes of statistical analysis and data processing are necessary before resubmitting the manuscript.

Major concerns:

1. In the pipeline of EEG preprocessing, the CAR can’t be done before the ICA. As warned by the tutorial of the EEGLab and the theory of ICA, the common averaged reference will reduce the rank of data. The sequence of preprocessing processes needs to be carefully examined.

2. What is the reference for choosing the time interval from 300ms to the reaction time for peak amplitude extraction? Please provide a further explanation of the temporal analysis window.

3. Please give a further illustration of the “median split” of dividing participants into two groups. Moreover, as the split is performed “for difference in RT between GG and PH”, this approach will increase the type I error of the following statistical analysis. If such a split is necessary, the split based on objective conditions such as baseline conditions is more acceptable.

4. In the paradigm design, four tasks, GG, PG, PP, and PH, are set. Why are there no GP and GH tasks?

5. In addition, how is the order of the actual scene and the virtual scene determined? Why is the actual scene placed before the virtual scene? Or in other words, why use PG instead of GP?

6. There should be four kinds of experiments in Randomized Walking Conditions in Figure 2, and only 1 kind of GG is drawn in this figure.

7. The authors need to supplement the questionnaire used by SAM. Was this questionnaire filled out before and after the experiment or just after the experiment for each condition?

8. The authors divided the subjects into 2 groups according to RT. Please add the RT result of each subject.

9. The authors conducted a correlation analysis between SAM and RT, but this paper wants to study the relationship between Physiological Stress and P300. Why do you not perform a correlation analysis between SAM and the amplitude or power of P300 in the two groups of subjects?

10. It can be seen from Figure 6 that Pz is not the strongest in the EEG scalp map of Group 2, so is it reasonable to use only Pz to calculate the P300 amplitude in Figure 7?

11. There are many problems in the statistical test. First, a major problem is using the paired test for multi-level comparison. As mentioned in the manuscript, “the RT, SAM, P3 peak and frequency power band data were not normally distributed”, and some non-parameter methods like the Friedman test can be used to determine whether multiple conditions are significantly different. The paired test can not determine the effect of a factor with multi-level. This problem influences the statistical test of RT, SAM, ERP amplitude, and the comparison between two groups. Second, the correction of the p-value is ignored when multi-comparisons are made. The manuscript doesn’t give an illustration of the correction method when RT, SAM, and ERSP are compared among four conditions (i.e. GG, PG, PP, and PH). Third, the objects of statistical analysis are not uniform. In some analyses, the object is the mean value of each subject, while in other analyses, the object is the value of each trial. Please choose a uniform object and ensure that the object meets the assumptions of the statistical test method.

5. The specific details of ERSP need further explanation. For example, the method of doing time-frequency decomposition, the window length of ERSP, the baseline correction method of ERSP, and the epoch range of data used by ERSP are needed to determine the effectiveness of ERSP results.

Minor issues

1. On page8 line 299, there should be repeated measures ANOVA instead of “repeated measure ANOVA”.

6. PLOS authors have the option to publish the peer review history of their article (what does this mean?). If published, this will include your full peer review and any attached files.

Reviewer #1: No

Reviewer #2: No

Reviewer #3: No

---

## [Author Response · Author response to Decision Letter 0]

21 Nov 2022

Dear PLOS ONE Editorial team and reviewers,

Thank you for providing us with detailed and well-written reviews/comments to help improve our manuscript. My apologies for the delay in providing the revision; we required an extension due to health issues in the prior months. We've reviewed the comments from the editor and reviewers and revised many parts of the paper. We hope our revision has addressed the concerns and comments from the reviews. We've provided the labelled (highlighted revisions) and the unlabelled manuscript in the revision. In this document, we will respond to the specific points raised by the editor and reviewers and provide details on how the revision addresses the points.

Editor/Additional Requirements:

We have modified our latex template to the PLOSOne template link on this website.

2. Please change "female" or "male" to "woman" or "man" as appropriate, when used as a noun (see for instance https://apastyle.apa.org/style-grammar-guidelines/bias-free-language/gender).

We have modified the nouns to the suggested more appropriate wording.

Addressed in point 1.

"This work was supported in part by the Australian Research Council (ARC) under discovery grant DP210101093 and DP220100803. Research was also sponsored in part by the Australia Defence Innovation Hub under Contract No. P18-650825, US Office of Naval Research Global under Cooperative Agreement Number ONRG - NICOP - N62909-19-1-2058, and AFOSR – DST Australian Autonomy Initiative agreement ID10134. We also thank the NSW Defence Innovation Network and NSW State Government of Australia for financial support in part of this research through grant DINPP2019 S1-03/09 and PP21-22.03.02."

"This work was supported by Australian Research Council (ARC, https://www.arc.gov.au/) (DP210101093 and DP220100803), Australia Defence Innovation Hub (https://innovationhub.defence.gov.au/, P18-650825), US Office of Naval Research Global (https://www.nre.navy.mil/, ONRG - NICOP - N62909-19-1-2058), AFOSR – DST Australia Autonomy Initiative (ID10134), and NSW Defence Innovation Network (https://innovationhub.defence.gov.au/, DINPP2019 S1-03/09 and PP21-22.03.02) to Chin-Teng Lin. The funders had no role in study design, data collection and analysis, decision to publish, or preparation of the manuscript."

My apologies for the oversight in the listing of funding bodies. We have removed the funding information from the manuscript. This funding statement listed above looks correct and won't need to be updated:

"This work was supported by Australian Research Council (ARC, https://www.arc.gov.au/) (DP210101093 and DP220100803), Australia Defence Innovation Hub (https://innovationhub.defence.gov.au/, P18-650825), US Office of Naval Research Global (https://www.nre.navy.mil/, ONRG - NICOP - N62909-19-1-2058), AFOSR – DST Australia Autonomy Initiative (ID10134), and NSW Defence Innovation Network (https://innovationhub.defence.gov.au/, DINPP2019 S1-03/09 and PP21-22.03.02) to Chin-Teng Lin. The funders had no role in study design, data collection and analysis, decision to publish, or preparation of the manuscript."

Upon re-submitting your revised manuscript, please upload your study's minimal underlying data set as either Supporting Information files or to a stable, public repository and include the relevant URLs, DOIs, or accession numbers within your revised cover letter. For a list of acceptable repositories, please see http://journals.plos.org/plosone/s/data-availability#loc-recommended-repositories. Any potentially identifying patient information must be fully anonymized.

We have uploaded the data required to replicate the results outlined in the manuscript to a public repository (Dryad). Please let us know if there is any issue with accessing the data. 

Repository link: https://datadryad.org/stash/share/tRM7LgrL_ecB9ghA632P_FXbMIYN6Yo8121G1f36mak

6. Please include your full ethics statement in the 'Methods' section of your manuscript file. In your statement, please include the full name of the IRB or ethics committee who approved or waived your study, as well as whether or not you obtained informed written or verbal consent. If consent was waived for your study, please include this information in your statement as well. 

We have updated the participant section in the methods to include the human research ethics committee (UTS), the approval ID, and the method of consent provided (informed and written). 

Reviewer 1:

Reviewer #1: The experimental setup is interesting. However, this reviewer has some concerns which have to address during major revision.

Thank you for the comments and the interesting paper suggestions; we have learnt a lot from the papers and included the methodology in our analysis and manuscript; we've cited the paper in the areas where references are from the suggested papers. 

1. The author should define "stress" in this context clearly. It would be better to replace "stress" by the word which can be measured quantitatively and/or having the standard questionnaire in identifying the level of mental or cognitive world load during interested periods. This famous related review article might help P. Consumer Grade EEG Measuring Sensors as Research Tools: A Review.

Thank you for pointing out the ambiguity in the definition of stress. We've added further clarification to the definition of physiological stress in the introduction (opening sentences). Specifically, we've outlined the Russell circumplex model of assessing stress through arousal and valence, which directly links to the SAM questionnaire used in this manuscript. These measures were also mentioned in the review article, so we included them in the introduction. 

We are hesitant to replace the word "stress" in this manuscript because multiple past similar works outlined in Table 1 directly use the word stress and relate it to the P300 response. Since this work aims to further the previous works listed directly, it would be more appropriate for the manuscript to follow the terminology and methodology established in the previous literature.

2. P300 or ERSP visualization and analysis need further improvement and exploration. The recent work with patients might help the author in the improvement of this point. This reviewer suggest the author to study more from A Pilot Study on Visually Stimulated Cognitive Tasks for EEG-Based Dementia Recognition.

The suggested paper is very relevant and helpful in revising the P300 and ERSP results. We've reworked our data analysis, and figures 8, 10, and 11 now follow closely with the methodology and presentation of the suggested paper. We've referenced the paper in our methodology to indicate where we followed the outlined method.

3. Also the analysis of EEG to together with response or reaction times can learn from: Supervision of a self-driving vehicle unmasks latent sleepiness relative to manually controlled driving.

This suggested paper was very helpful in improving our reaction time results. We followed the reaction time analysis outlined in the paper and incorporated figure 3 to show better the distribution of the participant's change in reaction time.

Looking forward to reading the revised version.

Thank you for the great paper suggestions; we hope we have appropriately addressed the concerns from the review.

Reviewer 2

Reviewer #2: This is an interesting study on P300 responses under real and virtual height exposure. Basically, the experimental design is somewhat unbalanced (and hard to understand from the description) but it's a good approach towards a quite timely question. And this is the main critique from my side. This is a nice pilot for a completely controlled study, but, in particular with some minor faults in the design and a sample size of 17 that additionally is divided into two groups, this is not sufficient for a publication. Along with this criticism, it should be noted that the main effect (the modulation of P3 amplitude in one of the two groups) just touches significance (Z=1.96, p=0.049) although the authors seem to have renounced to correct their data against alpha inflation.

As the authors claim in the final section, they were kind of surprised by the interindividual differences in behavior. 

Thank you for the detailed comments in the review; we highly appreciate the effort and thought into the feedback. We've revised our data analysis methodology and results to address your concerns with the manuscript, hopefully. Specifically, with the significance, we've revised the test to include post hoc analysis to produce more rigorous results.

Those are, besides the fact that mobile EEG is used in order to investigate more realistic behavior, an interesting point of this study, however, the authors nowhere provide evidence that there are really two distinct groups. They simply divided their sample by a split half on one RT parameter, which was for sure not bimodally distributed. The reader might have the impression that they searched for significance based on different analyses strategies and this one turned out to provide a p value below .05.

We've revised the division section to clarify the motivation behind splitting our sample group. We've also added figure 3, which better depicts the spread of difference in performance between the ground and heights condition. The main reason we chose this categorization is that (as shown in figure 3) is that some participants performed better in the high-stress condition (hence the negative difference between PH and GG). Hopefully, the revised text and figure 3 can clarify why we split the sample.

The entire MS appears to be intended in another context as presented. The introduction tlks a lot about BCIs, which are nowhere touched experimentally. This topic does not return before the final conclusion. The entire discussion section focuses on stress and cognition, which in my mind is the central issue here.

My apologies for the confusion/ambiguity with the mention of BCI. The inclusion of BCI was mainly due to differing opinions among the authors, as we wanted to include BCI as a higher-level motivation. We fully agree with your point that stress and cognition are the project's primary focus and key aim (investigating the effects of stress on cognitive performance). We've modified the introduction and methodology to be better focused on stress and cognition rather than BCI.

Minors

• The P300 paradigm is poorly described and not a standard for the P300 literature. Maybe this approach is usual in connection with PCIs, but the majority of visual oddball tasks uses much shorter stimulus presentations, longer interstimulus intervals which are normally additionally jittered.

Thank you for pointing out this issue; we have added more literature and rationale to the section. The trial structure contained elements of rapid serial visual presentation (rsvp) task design which is common for BCI research. We kept it named an oddball task mainly due to the timing and structure being closer to an oddball than an rsvp (as the stimuli are not rapidly presented). Please let us know if you feel the name should be changed.

• Why was the SAM presented before the trials? Would not a survey after performing the task better reflect stress?

We mainly chose to show the SAM before the start of the trials and during rests to minimize delay during the walking trials (participants would verbally rate the SAM score after each trial/walk). In hindsight, it would have been better to have visualized the SAM within the VR.

• Removing any component that is less than 85% classification for brain component (done with ICLabel?), is a hard criterion. How many ICs remained in the data then?

Thank you for pointing out this issue; we have revised our methodology, as 85% was a high criterion. We revised the preprocessing based on Makoto's pipeline (from sccn). The revised process considers the brain classification as the dominant label rather than a thresholded percentage. We also check for residual variance and dipole location.

• The authors appear to make separate tests for within group effecs and between groups effects on the same parameter. Was any correction against alpha error inflation applied?

We've revised the statistical test to now test for group effects. We applied a Friedman test (suggested by R3) and post hoc analysis through the Tukey HSD.

• statistical parameters are missing in many places. Also not significant effects should be presented with their numbers.

Thank you for pointing out this issue; we have revised the statistics and ensured consistent reporting. We've also added parameter reporting for when no significant effect was observed. Please let us know if we missed any specific parameter/section.

• Another evidence for effect searching: In the introduction they speak about Alpha and Theta ("276 pected P300 ERSP response is the alpha and theta synchronization during the onset (before) of the P300 peak and a gradual beta desynchronization during the P300 peak"), in the result section they present an effect in Beta power, and in the conclusion they added gamma activity. When I look at the time frequency plots, I rather see Mu responses in Group 2, which might reflect encapsulation due to the balance task.

My apologies for the confusing text, that line in the methodology refers to the background literature, which describes the characteristics of a P300 response in the ERSP. The results/conclusion reported on the findings from the data. We've fixed the section by adding the appropriate references to clarify the statement drawn from the literature. 

Reviewer 3:

Reviewer #3: This study tries to address the influence of stress under a naturalistic environment on the P300 response. This research is very practical, and the paradigm design is also very interesting. However, many critical problems of statistical analysis in this manuscript make the results of this study unconvincing. Carefully checking the processes of statistical analysis and data processing are necessary before resubmitting the manuscript.

Thank you for the thorough review and detailed comments; we've used your feedback to try to fix and revise the statistical analysis and reporting of the methodology. 

Major concerns:

1. In the pipeline of EEG preprocessing, the CAR can't be done before the ICA. As warned by the tutorial of the EEGLab and the theory of ICA, the common averaged reference will reduce the rank of data. The sequence of preprocessing processes needs to be carefully examined.

We agree that performing CAR reduces the data's rank and can be an issue if the number of ranks is equal to the number of channels. We should have (my apologies) mentioned that our EEG system used a reference electrode during the recording. So, the number of channels is 64 active electrodes + 1 FCz reference electrode. Before applying CAR, we add the reference electrode to the data. This action is equivalent to the "Add current reference channel back to the data "in the re-reference function by EEGLab. This way, when CAR is applied, we would have a rank of 64 equal to the number of probe electrodes and IC components after applying ICA. The code used below:

This issue was outlined in Makoto Miyakoshi's blog/forum site: https://sccn.ucsd.edu/wiki/Makoto's_preprocessing_pipeline

In -> 8. Re-reference the data to average -> What is rank?

Do, T. T. N., Lin, C. T., & Gramann, K. (2021). Human brain dynamics in active spatial navigation. Scientific Reports, 11(1), 1-12.

We have revised the preprocessing to clarify this issue, and the reference electrode was added back to avoid the rank issue.

2. What is the reference for choosing the time interval from 300ms to the reaction time for peak amplitude extraction? Please provide a further explanation of the temporal analysis window.

For the P300 peak detection, we chose the window 300-500ms based on the literature/previous papers on P300 peak detection. We've revised the methodology to clarify better why we chose the temporal analysis window and cite the reference we used.

3. Please give a further illustration of the "median split" of dividing participants into two groups. Moreover, as the split is performed "for difference in RT between GG and PH", this approach will increase the type I error of the following statistical analysis. If such a split is necessary, the split based on objective conditions such as baseline conditions is more acceptable.

We've revised the section to clarify better how and why we performed the median split. We also provided a figure that shows the individual participant's RT result (as requested in point 8).

4. In the paradigm design, four tasks, GG, PG, PP, and PH, are set. Why are there no GP and GH tasks?

The main reason is time, we originally had all 6 conditions, but our pilot experiments were around 3-4 hours. Each condition takes around 20-30 minutes. Our setup was quite complex as we had many safety protocols for walking on the platform, as it involved a safety harness and had fall risks. It would not have been feasible to perform all 6 conditions in a single session, and most participants were unwilling to come to multiple sessions (will need to repeat the setup). We chose to include GG, PP, and PH as the intuitive choice for three levels of height. The main choice was between PG, GP, and GH. The main novelty of the experiment was the walking platform, so we chose PG to explore the walking platform further. We reasoned that GP and GH were well documented in the literature, while we have not seen any condition where a person is physically elevated but virtually on the ground.

5. In addition, how is the order of the actual scene and the virtual scene determined? Why is the actual scene placed before the virtual scene? Or in other words, why use PG instead of GP?

My apologies if I misunderstand this question; the participant is wearing VR for all the conditions, so the virtual and actual scenes are played simultaneously, and the environment is mapped through optical motion capture.

If the question is why we chose PG over GP, it is mainly due to the reason mentioned in point 4: the physical platform was the main novelty. We wanted to include more conditions on the physical platform.

6. There should be four kinds of experiments in Randomized Walking Conditions in Figure 2, and only 1 kind of GG is drawn in this figure.

My apologies for the oversight in figure 2. Thank you for pointing this out. We've corrected the figure to include the four conditions. 

7. The authors need to supplement the questionnaire used by SAM. Was this questionnaire filled out before and after the experiment or just after the experiment for each condition?

Thank you for the reminder; we will attach the questionnaire sheet to the revision submission. The participants verbally provided a SAM score after each walk.

8. The authors divided the subjects into 2 groups according to RT. Please add the RT result of each subject.

Based on your suggestion, we have added Figure 3, which presents the individual RT results to clarify how better/why we split the sample.

9. The authors conducted a correlation analysis between SAM and RT, but this paper wants to study the relationship between Physiological Stress and P300. Why do you not perform a correlation analysis between SAM and the amplitude or power of P300 in the two groups of subjects?

Thank you for the suggestion; based on this, we added Figure 9 and an additional result section for the correlation between SAM and P300 peak amplitude.

10. It can be seen from Figure 6 that Pz is not the strongest in the EEG scalp map of Group 2, so is it reasonable to use only Pz to calculate the P300 amplitude in Figure 7?

We've revised the topography results based on EEG analysis methodology changes. The new topography result seems consistent with Pz being within the activation region. We also want to keep consistent past stress, and P300 works referenced in table 1 feature Pz in their analysis.

11. There are many problems in the statistical test. First, a major problem is using the paired test for multi-level comparison. As mentioned in the manuscript, "the RT, SAM, P3 peak and frequency power band data were not normally distributed", and some non-parameter methods like the Friedman test can be used to determine whether multiple conditions are significantly different. The paired test can not determine the effect of a factor with multi-level. This problem influences the statistical test of RT, SAM, ERP amplitude, and the comparison between two groups. Second, the correction of the p-value is ignored when multi-comparisons are made. The manuscript doesn't give an illustration of the correction method when RT, SAM, and ERSP are compared among four conditions (i.e. GG, PG, PP, and PH). 

Based on your feedback, we've revised the statistical analysis in the methodology to the Friedman Test with Post Hoc Tukey HSD for condition comparison. 

Third, the objects of statistical analysis are not uniform. In some analyses, the object is the mean value of each subject, while in other analyses, the object is the value of each trial. Please choose a uniform object and ensure that the object meets the assumptions of the statistical test method.

We've also revised the results by removing the trial/repeated measured analysis and having all the measurements uniform to the mean value by subject.

5. The specific details of ERSP need further explanation. For example, the method of doing time-frequency decomposition, the window length of ERSP, the baseline correction method of ERSP, and the epoch range of data used by ERSP are needed to determine the effectiveness of ERSP results.

We've revised the ERSP section to specify the steps involved in feature extraction, as outlined in this point.

Minor issues

1. On page8 line 299, there should be repeated measures ANOVA instead of "repeated measure ANOVA".

This line was removed after the revision to the statistical method. No longer using the repeated measure.

Thank you for your feedback; please let me know if you have any additional comments or feedback.

Kind regards,

Howe

---

## [Decision Letter · Decision Letter 1]

7 Mar 2023

PONE-D-22-20226R1Understanding the Effects of Physiological Stress on the P300 ResponsePLOS ONE

Dear Dr. Zhu,

Thank you for submitting your manuscript to PLOS ONE. After careful consideration, we feel that it has merit but does not fully meet PLOS ONE’s publication criteria as it currently stands. Therefore, we invite you to submit a revised version of the manuscript that addresses the points raised during the review process.

We look forward to receiving your revised manuscript.

Kind regards,

Steve Zimmerman, PhD

Associate Editor, PLOS ONE

Additional Editor Comments: 

Thank you for your careful revision to your manuscript. Unfortunately two of the three original reviewers were unable to review your revised submission, therefore we sought out one additional reviewer. This reviewer (reviewer 4) still has some concerns about the the definition of physiological stress used, and the differentiation between groups (see attached file).

Could you please carefully revise the manuscript to address all comments raised?

Reviewers' comments:

Reviewer's Responses to Questions

**Comments to the Author**

1. If the authors have adequately addressed your comments raised in a previous round of review and you feel that this manuscript is now acceptable for publication, you may indicate that here to bypass the “Comments to the Author” section, enter your conflict of interest statement in the “Confidential to Editor” section, and submit your "Accept" recommendation.

Reviewer #1: All comments have been addressed

Reviewer #4: (No Response)

2. Is the manuscript technically sound, and do the data support the conclusions?

Reviewer #1: Yes

Reviewer #4: Partly

3. Has the statistical analysis been performed appropriately and rigorously? 

Reviewer #1: Yes

Reviewer #4: Yes

4. Have the authors made all data underlying the findings in their manuscript fully available?

Reviewer #1: Yes

Reviewer #4: Yes

5. Is the manuscript presented in an intelligible fashion and written in standard English?

Reviewer #1: Yes

Reviewer #4: Yes

6. Review Comments to the Author

Reviewer #1: (No Response)

Reviewer #4: More information is required to qualify claims made concerning the measures taken linking to the definition of stress used (including in the title). The study is a sound basis as a pilot for a more comprehensive study addressing points made in attached reviewer feedback about the differentiation of groups on which the main conclusions are based.

7. PLOS authors have the option to publish the peer review history of their article (what does this mean?). If published, this will include your full peer review and any attached files.

Reviewer #1: No

Reviewer #4: No

---

## [Author Response · Author response to Decision Letter 1]

25 May 2023

Thank you for the detailed review from the reviewers and editors. I would like to apologise for the late response and revision. The extension was requested primarily due to health issues which delayed the revisions. We hope our revisions and response have addressed the concerns of the reviewers. 

Reviewer 1

Thank you for the previous feedback and recommendations for improving our paper. We appreciate the time and thought spent on the reviewing process.

Reviewer 4 (more detailed notes from the response document)

Thank you for the comprehensive feedback and clear suggestions. Based on your recommendation, I have revised the definition of stress in the manuscript by removing the physiological stress component and focusing around stress elicitation. I have revised the text to better clarify arousal and how the SAM arousal component relates to stress. As suggested, I have also adjusted the methodology to better clarify decisions, steps, and additional information such as trial distribution. The response document better highlights the points of your feedback. I hope this revision and the response has adequately addressed your concerns and suggestions.

---

## [Decision Letter · Decision Letter 2]

27 Jul 2023

PONE-D-22-20226R2Understanding the Effects of Stress on the P300 ResponsePLOS ONE

Dear Dr. Zhu,

Thank you for submitting your manuscript to PLOS ONE. After careful consideration, we feel that it has merit but does not fully meet PLOS ONE’s publication criteria as it currently stands. Therefore, we invite you to submit a revised version of the manuscript that addresses the points raised during the review process.

We look forward to receiving your revised manuscript.

Kind regards,

Valentina Bruno

Academic Editor

PLOS ONE

Journal Requirements:

Reviewers' comments:

Reviewer's Responses to Questions

**Comments to the Author**

1. If the authors have adequately addressed your comments raised in a previous round of review and you feel that this manuscript is now acceptable for publication, you may indicate that here to bypass the “Comments to the Author” section, enter your conflict of interest statement in the “Confidential to Editor” section, and submit your "Accept" recommendation.

Reviewer #4: (No Response)

2. Is the manuscript technically sound, and do the data support the conclusions?

Reviewer #4: Partly

3. Has the statistical analysis been performed appropriately and rigorously? 

Reviewer #4: Yes

4. Have the authors made all data underlying the findings in their manuscript fully available?

Reviewer #4: Yes

5. Is the manuscript presented in an intelligible fashion and written in standard English?

Reviewer #4: Yes

6. Review Comments to the Author

Reviewer #4: The authors have resubmitted the manuscript: “Understanding the Effects of Stress on the P300 Response” following reviewer feedback that raised some concerns based on the previous draft for submission. A primary issue highlighted previously centred on the title and contention that the study was evaluating physiological aspects of stress in the experimental scenario employed. Questions were also raised in relation to the Self Assessment Mannikin measures used: specifically only measuring the arousal dimensions (and not the valence and dominance components of the SAM). Objections were raised as to whether the experimental set-up was investigating physiological or psychological aspects of stress, and that by only capturing data on the arousal dimension, this might provide an incomplete picture of the model of stress being alluded to. Further comments questioned claims regarding ability to cope with stress and P300, despite no effects being observed when groups were split.

The authors have addressed key points including specific allusions to ‘physiological’ stress, removing reference to this. They also explained that the logistics of the experimental scenario meant that incorporating all of the dimensions of the SAM would have interfered with the naturalistic context in which the participant was situated. They provide a rationale on why the arousal dimension is valid to include as a measure in itself, as this has been shown to have a close relationship to threat perception. This reviewer recognizes the unique challenge that exists when running experiments that preserve a naturalistic context in which to elicit behaviours of interest that occur in real world situations, whilst also serving to employ robust measures that are usually collected in more contrived laboratory scenarios. This addresses the concerns around definitions of stress and the reasoning for using the subjective measures (SAM).

However, with respect to the analysis and assertions made about how the P300 relates to ability to cope with stress, the authors were unable to split participants based on a marker of stress. Rather, the split between groups was done on the basis of reaction times. As mentioned previously, splitting by reaction times can involve confounds unrelated to stress. This should be acknowledged in the manuscript, or risk overstating claims made. Whilst the authors now provide the distribution of trials, it remains unclear if any speed-accuracy trade-off was apparent, or if there were any statistical differences between groups regarding error rates.

On the basis of the points addressed but with respect to the basis for splitting the data, it is recommended that minor revisions of the manuscript are made, modifying the discussion and conclusions to reflect the possibility that differences due to the split on RTs may not be specifically due to stress, and downplaying claims made about markers of ability to cope with stress. Such minor revisions would in this reviewer’s view satisfy criteria for publication.

One additional suggestion concerns the title. It may be useful to emphasise the nature of the stressor/scenario from which the effects of stress are being derived, for specificity. For instance highlighting that this work is seeking to novelly extend research into the stress response/P300 into a more naturalistic environmental context, which is a particular strength of the work. This could prove fruitful for future lines of enquiry that differentiate ‘stress’ according to varied contexts beyond height exposure.

7. PLOS authors have the option to publish the peer review history of their article (what does this mean?). If published, this will include your full peer review and any attached files.

Reviewer #4: No

---

## [Author Response · Author response to Decision Letter 2]

7 Aug 2023

Reviewer 4

Response: We highly appreciate the time and effort taken to review our manuscript. Thank you for recognising the strengths and weaknesses of this manuscript and providing clear suggestions for improving this manuscript. We hope our revisions and response have addressed the concerns of the reviewers. 

The authors have resubmitted the manuscript: “Understanding the Effects of Stress on the P300 Response” following reviewer feedback that raised some concerns based on the previous draft for submission. A primary issue highlighted previously centred on the title and contention that the study was evaluating physiological aspects of stress in the experimental scenario employed. Questions were also raised in relation to the Self Assessment Mannikin measures used: specifically only measuring the arousal dimensions (and not the valence and dominance components of the SAM). Objections were raised as to whether the experimental set-up was investigating physiological or psychological aspects of stress, and that by only capturing data on the arousal dimension, this might provide an incomplete picture of the model of stress being alluded to. Further comments questioned claims regarding ability to cope with stress and P300, despite no effects being observed when groups were split.

The authors have addressed key points including specific allusions to ‘physiological’ stress, removing reference to this. They also explained that the logistics of the experimental scenario meant that incorporating all of the dimensions of the SAM would have interfered with the naturalistic context in which the participant was situated. They provide a rationale on why the arousal dimension is valid to include as a measure in itself, as this has been shown to have a close relationship to threat perception. This reviewer recognizes the unique challenge that exists when running experiments that preserve a naturalistic context in which to elicit behaviours of interest that occur in real world situations, whilst also serving to employ robust measures that are usually collected in more contrived laboratory scenarios. This addresses the concerns around definitions of stress and the reasoning for using the subjective measures (SAM).

However, with respect to the analysis and assertions made about how the P300 relates to ability to cope with stress, the authors were unable to split participants based on a marker of stress. Rather, the split between groups was done on the basis of reaction times. As mentioned previously, splitting by reaction times can involve confounds unrelated to stress. This should be acknowledged in the manuscript, or risk overstating claims made. Whilst the authors now provide the distribution of trials, it remains unclear if any speed-accuracy trade-off was apparent, or if there were any statistical differences between groups regarding error rates.

On the basis of the points addressed but with respect to the basis for splitting the data, it is recommended that minor revisions of the manuscript are made, modifying the discussion and conclusions to reflect the possibility that differences due to the split on RTs may not be specifically due to stress, and downplaying claims made about markers of ability to cope with stress. Such minor revisions would in this reviewer’s view satisfy criteria for publication.

Response:Thank you for your understanding and clear suggestion for the minor revision. We have revised the language to reflect better the claims made in this manuscript. 

-We have revised the methodology, discussion, and conclusion to outline the potential confounding factors of the splitting methodology. We highlighted that other factors may confound the results and reworded the claims to include the possibility that these factors can affect the results. 

-We have revised the abstract, introduction, discussion, and conclusion by rewording the statements on measuring the participant’s ability to cope with stress. As outlined by the reviewer, we agree that this claim would be difficult to justify with the measurements collected in this study. We have revised the text to reflect the more exploratory aspect of understanding the P300 response when stressed rather than devising a new metric.

One additional suggestion concerns the title. It may be useful to emphasise the nature of the stressor/scenario from which the effects of stress are being derived, for specificity. For instance highlighting that this work is seeking to novelly extend research into the stress response/P300 into a more naturalistic environmental context, which is a particular strength of the work. This could prove fruitful for future lines of enquiry that differentiate ‘stress’ according to varied contexts beyond height exposure.

Response:Thank you for this suggestion; we agree and have adjusted the title to include the aspects of naturalistic height exposure.

---

## [Decision Letter · Decision Letter 3]

5 Dec 2023

PONE-D-22-20226R3Understanding the Effects of Stress on the P300 Response During Naturalistic Simulation of Heights ExposurePLOS ONE

Dear Dr. Zhu,

Thank you for submitting your manuscript to PLOS ONE. After careful consideration, we feel that it has merit but does not fully meet PLOS ONE’s publication criteria as it currently stands. Therefore, we invite you to submit a revised version of the manuscript that addresses the points raised during the review process.

We look forward to receiving your revised manuscript.

Kind regards,

Valentina Bruno

Academic Editor

PLOS ONE

Journal Requirements:

Reviewers' comments:

Reviewer's Responses to Questions

**Comments to the Author**

1. If the authors have adequately addressed your comments raised in a previous round of review and you feel that this manuscript is now acceptable for publication, you may indicate that here to bypass the “Comments to the Author” section, enter your conflict of interest statement in the “Confidential to Editor” section, and submit your "Accept" recommendation.

Reviewer #2: (No Response)

Reviewer #4: All comments have been addressed

Reviewer #5: All comments have been addressed

2. Is the manuscript technically sound, and do the data support the conclusions?

Reviewer #2: (No Response)

Reviewer #4: Yes

Reviewer #5: Yes

3. Has the statistical analysis been performed appropriately and rigorously? 

Reviewer #2: (No Response)

Reviewer #4: Yes

Reviewer #5: Yes

4. Have the authors made all data underlying the findings in their manuscript fully available?

Reviewer #2: (No Response)

Reviewer #4: Yes

Reviewer #5: Yes

5. Is the manuscript presented in an intelligible fashion and written in standard English?

Reviewer #2: (No Response)

Reviewer #4: Yes

Reviewer #5: Yes

6. Review Comments to the Author

Reviewer #2: (No Response)

Reviewer #4: The authors have resubmitted the manuscript: “Understanding the Effects of Stress on the P300 Response During Naturalistic Simulation of Heights Exposure” following additional reviewer comments recommending minor revision to the previous revised draft.

The reviewer feedback concerning the split between groups has been more or less addressed with respect to potential confounds and limitations (eg. RTs perhaps being affected by factors other than ‘stress’ per se), including a reference to the accuracy scores. Although in the limitations section there is emphasis placed on RTs with respect to utility as a metric for gauging/estimating susceptibility to stress in conjunction with brain behaviour.

Other comments have been addressed from earlier revisions concerning definition of stress, although this reviewer still has some reservations about the qualification of stress in the experiment.

“One group performed worse when stressed” - statements like this make presumption that a stress response has been elicited rather than the contention that the cognitive-affective-arousal state markers may be indicative of so-called ‘stress’ - this may sound like splitting hairs but nevertheless feels poignant to the argument being made in the paper about potential indicators of stress identified via the P300 and cognitive-arousal markers of performance...Indeed, the authors refer to ‘stress levels’ when they are talking about the different conditions. These would perhaps be more objectively described as ‘potential stress-inducing conditions’ - isn’t the point of the experiment to examine whether, and to what degree, these conditions do actually elicit changes in ‘stress levels’?

Another point of contention rests in the defining of stress, in absolute, as a negatively valenced state. Whilst the authors reference the circumplex model, with stress generally associated with high arousal/negative valence, this generalisation does not reveal the whole picture. Further subdivision of stress into negative and positive dimensions (as is increasingly occurring in the literature – eg. Bak et al., 2022) - ‘distress’ and ‘eustress’ - paints a more complex and varied picture of stress. This may ultimately contribute to resolution of inconsistency in findings in the literature with respect for instance, to alternately, cognition-impairing and cognition-enhancing effects of exposure to ‘stress’. It is a relevant point, particularly as research extends further into more naturalistic experimental scenarios territory, as the current study valiantly attempts. This is alluded to somewhat in the section: “Understanding Yerkes-Dodson Law and Participant’s Performance when Stressed” in discussion of Group 2 vs Group 1 experiencing ‘more optimal stress level’.

[Bak, S., Shin, J., Jeong, J. (2022). Subdividing Stress Groups into Eustress and Distress Groups Using Laterality Index Calculated from Brain Hemodynamic Response. Biosensors (Basel). 12(1):33.]

“In summary, the two groups we found exhibited seemingly contradictory yet rational and supported by literature, P300 behaviours” - this sentence doesn’t read clearly.

In this reviewer’s opinion the points raised from previous review have been somewhat satisfactorily addressed, but note the above comments with respect to wider debate about what constitutes ‘stress’ and how the label is employed with respect to outlining the experimental conditions and asserting that a state of stress is variably elicited.

On the basis of the revisions made this reviewer recommends that the paper is fit for submission.

Reviewer #5: This is a very interesting study, will contribute much for understanding the neural correlates of Stress. I appreciate this work on the idea and manipulations in the experiments, and also have some suggestions for clear state the findings and implications.

1. How to clearly distinguish the effect of emotional and stress during the procedure of experiment?

2. Is it possible to consider the link between stress and neural response in line with a curve model? That is, why participants performed better in some specific not all the conditions? How to reasonably to explain it?

3. What's the meanness to discriminate the real and virtual environment?

4. Did you control the daily stress and stress sensibilities in the data analysis?

7. PLOS authors have the option to publish the peer review history of their article (what does this mean?). If published, this will include your full peer review and any attached files.

Reviewer #2: No

Reviewer #4: No

Reviewer #5: **Yes: **yes

---

## [Author Response · Author response to Decision Letter 3]

18 Jan 2024

Happy New Year! Thank you for the time the reviewers and editors took to review and provide feedback to improve this manuscript. We hope our revisions and response have addressed the concerns of the reviewers. 

Reviewer 4

The reviewer feedback concerning the split between groups has been more or less addressed with respect to potential confounds and limitations (eg. RTs perhaps being affected by factors other than ‘stress’ per se), including a reference to the accuracy scores. Although in the limitations section there is emphasis placed on RTs with respect to utility as a metric for gauging/estimating susceptibility to stress in conjunction with brain behaviour.

Other comments have been addressed from earlier revisions concerning definition of stress, although this reviewer still has some reservations about the qualification of stress in the experiment.

Response: We highly appreciate the time and effort taken to review our manuscript. Thank you for providing clear suggestions for improving this manuscript. We hope our revisions and response have addressed the concerns of the reviewers. 

“One group performed worse when stressed” - statements like this make presumption that a stress response has been elicited rather than the contention that the cognitive-affective-arousal state markers may be indicative of so-called ‘stress’ - this may sound like splitting hairs but nevertheless feels poignant to the argument being made in the paper about potential indicators of stress identified via the P300 and cognitive-arousal markers of performance...Indeed, the authors refer to ‘stress levels’ when they are talking about the different conditions. These would perhaps be more objectively described as ‘potential stress-inducing conditions’ - isn’t the point of the experiment to examine whether, and to what degree, these conditions do actually elicit changes in ‘stress levels’?

Thank you for highlighting your concern about using ‘stress level’ in this paper. We’ve reworded the paper to clarify the terming of stress better. The primary purpose of the experiment is to assess the behaviour of the P300 response when under stress. Walking at heights or height exposure is a recently popular stress elicitation paradigm, e.g. Peterson et al. (published in PLOS ONE). Based on our understanding and observation in the literature about height exposure and beam walking paradigms, stress is the most appropriate terminology for the paper. 

Peterson, S.M., Furuichi, E. and Ferris, D.P., 2018. Effects of virtual reality high heights exposure during beam-walking on physiological stress and cognitive loading. PloS one, 13(7), p.e0200306.

Another point of contention rests in the defining of stress, in absolute, as a negatively valenced state. Whilst the authors reference the circumplex model, with stress generally associated with high arousal/negative valence, this generalisation does not reveal the whole picture. Further subdivision of stress into negative and positive dimensions (as is increasingly occurring in the literature – eg. Bak et al., 2022) - ‘distress’ and ‘eustress’ - paints a more complex and varied picture of stress. This may ultimately contribute to resolution of inconsistency in findings in the literature with respect for instance, to alternately, cognition-impairing and cognition-enhancing effects of exposure to ‘stress’. It is a relevant point, particularly as research extends further into more naturalistic experimental scenarios territory, as the current study valiantly attempts. This is alluded to somewhat in the section: “Understanding Yerkes-Dodson Law and Participant’s Performance when Stressed” in discussion of Group 2 vs Group 1 experiencing ‘more optimal stress level’.

[Bak, S., Shin, J., Jeong, J. (2022). Subdividing Stress Groups into Eustress and Distress Groups Using Laterality Index Calculated from Brain Hemodynamic Response. Biosensors (Basel). 12(1):33.]

Response: We agree that having valance measurements would have provided some interesting findings and potential to differentiate the participant experience further. Thank you for highlighting this interesting paper; we have revised the paper to include this work in our discussion. 

“In summary, the two groups we found exhibited seemingly contradictory yet rational and supported by literature, P300 behaviours” - this sentence doesn’t read clearly.

We’ve revised this sentence to improve the clarity. 

Reviewer 5

 This is a very interesting study, will contribute much for understanding the neural correlates of Stress. I appreciate this work on the idea and manipulations in the experiments, and also have some suggestions for clear state the findings and implications.

Response: Thank you for your time and feedback. We hope that we have sufficiently answered your questions.

1. How to clearly distinguish the effect of emotional and stress during the procedure of experiment?

Response: For the experimental paradigm, we are only considering the increase in arousal as the main indicator of stress. The emotional factor could play a small role in the response. However, the height exposure paradigm is primarily designed to evoke a physiological stress response. As observed in our SAM results, the paradigm was able to heighten the participant’s arousal levels.

2. Is it possible to consider the link between stress and neural response in line with a curve model? That is, why participants performed better in some specific not all the conditions? How to reasonably to explain it?

Response: Exploring a polynomial regression model could be interesting. In our discussion, we explored the Yorkes-Dodson model, a bell curve. Unfortunately, our correlation results do not have sufficient data points to apply a 2nd or 3rd-order polynomial regression without overfitting. Hence, our results only used linear regression. 

Our discussions discuss potential reasons why certain participants performed better in some conditions. By splitting the participants, we observed a difference in EEG indicators, suggesting that certain participants were less able to focus when in the stressful conditions/

3. What's the meanness to discriminate the real and virtual environment?

Response: We’re not certain about the meaning of this question. Is there a specific section of the paper this question is referring to? 

4. Did you control the daily stress and stress sensibilities in the data analysis?

Response: We did not control for daily stress factors. Our work mainly follows the beam walking and height exposure virtual paradigms of Peterson et al. and Meehan et al. Exploring daily stress and stress susceptibility responses is less common in these exposure experiment designs. Height exposure is a reliable stress paradigm across various participant demographics. Short-term exposure can reliably elicit a response. This type of paradigm has become more popular among computational neuroscientists and behaviour psychological researchers due to the reliability and ease of set-up (due to Virtual reality).

Peterson, S.M., Furuichi, E. and Ferris, D.P., 2018. Effects of virtual reality high heights exposure during beam-walking on physiological stress and cognitive loading. PloS one, 13(7), p.e0200306.

Meehan, M., Insko, B., Whitton, M. and Brooks Jr, F.P., 2002. Physiological measures of presence in stressful virtual environments. Acm transactions on graphics (tog), 21(3), pp.645-652.

Examples of virtual height exposure paradigms 

Kisker, J., Gruber, T. and Schöne, B., 2021. Behavioral realism and lifelike psychophysiological responses in virtual reality by the example of a height exposure. Psychological research, 85, pp.68-81.

Fadeev, K.A., Smirnov, A.S., Zhigalova, O.P., Bazhina, P.S., Tumialis, A.V. and Golokhvast, K.S., 2020. Too real to be virtual: Autonomic and EEG responses to extreme stress scenarios in virtual reality. Behavioural neurology, 2020.

Krupić, D., Žuro, B. and Corr, P.J., 2021. Anxiety and threat magnification in subjective and physiological responses of fear of heights induced by virtual reality. Personality and Individual Differences, 169, p.109720.

Arachchige, S.N.K., Chander, H., Shojaei, A., Knight, A.C., Brown, C., Freeman, H.R. and Chen, C.C., 2024. Effects of virtual heights, dual-tasking, and training on static postural stability. Applied Ergonomics, 114, p.104145.

---

## [Decision Letter · Decision Letter 4]

20 Feb 2024

PONE-D-22-20226R4Understanding the Effects of Stress on the P300 Response During Naturalistic Simulation of Heights ExposurePLOS ONE

Dear Dr. Zhu,

Thank you for submitting your manuscript to PLOS ONE. After careful consideration, we feel that it has merit but does not fully meet PLOS ONE’s publication criteria as it currently stands. Therefore, we invite you to submit a revised version of the manuscript that addresses the points raised during the review process.

We look forward to receiving your revised manuscript.

Kind regards,

Valentina Bruno

Academic Editor

PLOS ONE

Journal Requirements:

Reviewers' comments:

Reviewer's Responses to Questions

**Comments to the Author**

1. If the authors have adequately addressed your comments raised in a previous round of review and you feel that this manuscript is now acceptable for publication, you may indicate that here to bypass the “Comments to the Author” section, enter your conflict of interest statement in the “Confidential to Editor” section, and submit your "Accept" recommendation.

Reviewer #5: All comments have been addressed

2. Is the manuscript technically sound, and do the data support the conclusions?

Reviewer #5: Partly

3. Has the statistical analysis been performed appropriately and rigorously? 

Reviewer #5: Yes

4. Have the authors made all data underlying the findings in their manuscript fully available?

Reviewer #5: Yes

5. Is the manuscript presented in an intelligible fashion and written in standard English?

Reviewer #5: Yes

6. Review Comments to the Author

Reviewer #5: Thanks for reviewing the paper again, the quality of the article has improved much. The authors answered the questions accurately and rigiously, I have one question that How to difine the virtual and real enviorment in the present study. How to exclude the effect of experiences from real environment in the virtual environment condition?

7. PLOS authors have the option to publish the peer review history of their article (what does this mean?). If published, this will include your full peer review and any attached files.

Reviewer #5: **Yes: **Fanchang Kong

---

## [Author Response · Author response to Decision Letter 4]

22 Feb 2024

Thank you for the time and effort you put into reviewing and significantly improving our manuscript. We hope our revisions and response have addressed the concerns of the reviewers. 

Reviewer 5

Thanks for reviewing the paper again, the quality of the article has improved much. The authors answered the questions accurately and rigiously, I have one question that How to difine the virtual and real enviorment in the present study. How to exclude the effect of experiences from real environment in the virtual environment condition?

Response: We highly appreciate your suggestions, which improved our manuscript throughout the revision process. 

In regards to the question, there are two instances concerning the virtual and physical/real-world environment. The first part is the local physical walking platform (Fig. 1.) in conjunction with the virtual environment. The second part is the paradigm being a more naturalistic real-world environment.

1. Physical and Virtual walking platform. 

In our study, the physical and virtual environments work together to deliver an immersive height experience. Our work follows quite closely with several publications by Meehan et al. [1-3]. These publications are quite well-known in virtual reality and cognitive neuroscience for exploring the psychological aspects of physical and virtual environments. Their work summarises that a better sense of presence and immersion results in a stronger stress elicitation as human can suspend their sense of disbelief. This can be achieved by creating enough sensory congruence (consistency of senses) in both the physical and virtual environment. 

In our case (outlined in Methodology lines 92-135), we generate visual and auditory stimuli in the virtual environment. The physical environment supplies the haptic feedback to give a sense of walking on a platform and elevation. 

“How to define the virtual and real environment in the present study.”

We have revised some of the wording in the paper to ensure that ‘physical’ environment is used to describe the local study set-up of the walking platform. The virtual environment is the display of the virtual world that the user operates in through the virtual reality headset. 

“How to exclude the effect of experiences from real environment in the virtual environment condition?”

In this context, we mainly focus on using the virtual and physical environment to immerse the user into suspending their sense of disbelief. The main motivation for this type of stimuli is based on the view suggested by Meehan et al. [2] that if the person has a strong sense of presence in the environment, they are more likely to generate a consistent stress response. 

When designing the study, the challenge wouldn’t necessarily be to exclude or remove the real-world factor. Rather, how can we give the person a strong sense of presence within the height-exposure environment to elicit a stress response? In our case, from previous literature and previous studies with this set-up, we found that this type of short-term elicitation can generate a strong and consistent stress response (observable in the ERP/ERSP).

[1] Meehan, M., Razzaque, S., Whitton, M.C. and Brooks, F.P., 2003, March. Effect of latency on presence in stressful virtual environments. In IEEE Virtual Reality, 2003. Proceedings. (pp. 141-148). IEEE.

[2] Meehan, M., Razzaque, S., Insko, B., Whitton, M. and Brooks, F.P., 2005. Review of four studies on the use of physiological reaction as a measure of presence in stressfulvirtual environments. Applied psychophysiology and biofeedback, 30, pp.239-258.

[3] Meehan, M., Insko, B., Whitton, M. and Brooks Jr, F.P., 2002. Physiological measures of presence in stressful virtual environments. Acm transactions on graphics (tog), 21(3), pp.645-652.

2. Real-world as a naturalistic environment

We’ve revised the paper to ensure that any “real-world” or “real world” usage refers to the naturalistic or in-the-wild environment. In this case, we describe a complex environment where the user will likely contend with more complex sensory experiences and different taskings. Our work argues that our paradigm provides a more complex sensory experience (visual, auditory, somatosensory, and haptic) and task design (walking and oddball tasks). Therefore, it is a more naturalistic stress elicitation method. This contrasts social or mental task paradigms for stress elicitation (outlined in Table 1). It could argued that these situations may appear in the real-world environment. Our view is that the traditional paradigms hold relatively low complexity compared to our design, which would be a better analogue translation study. Based on this, our analysis approach follows the MOBI (mobile brain imaging) method proposed by Klaus Gramman (one of our close colleagues), adapted for ambulatory study designs.

---

## [Decision Letter · Decision Letter 5]

11 Mar 2024

Understanding the Effects of Stress on the P300 Response During Naturalistic Simulation of Heights Exposure

PONE-D-22-20226R5

Dear Dr. Zhu,

We’re pleased to inform you that your manuscript has been judged scientifically suitable for publication and will be formally accepted for publication once it meets all outstanding technical requirements.

Kind regards,

Valentina Bruno

Academic Editor

PLOS ONE

Additional Editor Comments (optional):

Reviewers' comments:

Reviewer's Responses to Questions

**Comments to the Author**

1. If the authors have adequately addressed your comments raised in a previous round of review and you feel that this manuscript is now acceptable for publication, you may indicate that here to bypass the “Comments to the Author” section, enter your conflict of interest statement in the “Confidential to Editor” section, and submit your "Accept" recommendation.

Reviewer #5: All comments have been addressed

2. Is the manuscript technically sound, and do the data support the conclusions?

Reviewer #5: Yes

3. Has the statistical analysis been performed appropriately and rigorously? 

Reviewer #5: Yes

4. Have the authors made all data underlying the findings in their manuscript fully available?

Reviewer #5: Yes

5. Is the manuscript presented in an intelligible fashion and written in standard English?

Reviewer #5: Yes

6. Review Comments to the Author

Reviewer #5: The authors have answered my suggestions well, I think, I have no new questions for them. The current state of MS should be accepted to publish.

7. PLOS authors have the option to publish the peer review history of their article (what does this mean?). If published, this will include your full peer review and any attached files.

Reviewer #5: **Yes: **Fanchang Kong

---

## [Editor Report · Acceptance letter]

25 Mar 2024

PONE-D-22-20226R5 

PLOS ONE

Dear Dr. Zhu, 

I'm pleased to inform you that your manuscript has been deemed suitable for publication in PLOS ONE. Congratulations! Your manuscript is now being handed over to our production team.

Kind regards, 

on behalf of

Dr. Valentina Bruno 

Academic Editor

PLOS ONE